Subject Areas:
environmental science

Keywords:
organic residue analysis, beeswax, lipids, Chalcolithic, southern Levant, GC-MS

Author for correspondence:
Rivka Chasan
e-mail: rchasan@campus.haifa.ac.il

# Bee products in the prehistoric southern levant: evidence from the lipid organic record

Rivka Chasan[1], Danny Rosenberg[1], Florian Klimscha[2], Ron Beeri[3], Dor Golan[3], Ayelet Dayan[3], Ehud Galili[4,5] and Cynthianne Spiteri[6]

[1]Laboratory for Ground Stone Tools Research, Zinman Institute of Archaeology, University of Haifa, Haifa, Israel
[2]Archaeology Division, Research/Collections, Lower Saxony State Museum, Hanover, Germany
[3]Israel Antiquities Authority, Jerusalem, Israel
[4]Zinman Institute of Archaeology, and [5]Leon Recanati Institute for Maritime Studies, University of Haifa, Haifa, Israel
[6]Institute of Pre- and Protohistoric Archaeology, Eberhard Karls Universität Tübingen, Tübingen, Germany

 RC, 0000-0002-0150-9250; DR, 0000-0001-6991-4298;
CS, 0000-0003-1773-3895

Beehive products have a rich global history. In the wider Levantine region, bees had a significant role in Egypt and Mesopotamia, and intensive beekeeping was noted in Israel during the Biblical period when apiaries were first identified. This study investigates the origins of this extensive beekeeping through organic residue analysis of pottery from prehistoric sites in the southern Levant. The results suggest that beehive products from likely wild bees were used during the Chalcolithic period as a vessel surface treatment and/or as part of the diet. These functions are reinforced by comparison to the wider archaeological record. While the true frequency of beeswax use may be debated, alternatives to beehive products were seemingly preferred as wild resources contrasted with the socio-economic system centred on domesticated resources, controlled production and standardization. Bee products only became an important part of the economic canon in the southern Levant several millennia later.

## 1. Introduction

Bees produce a variety of products that can be readily exploited, including beeswax, honey and propolis, each having an extensive history of use [1]. Beeswax was used for several

purposes, including lighting [2], gluing [3], medicine [4–6], art [7,8], pottery sealing [9], embalming [10] and metallurgy [11]. Honey was frequently used as a sweetener and preservative, but it also played a role in medicine and embalming [1]. Propolis was similarly used as medicine [12] and as an adhesive [1].

The use of bee products may be identified in the archaeological record using varying lines of data including textual sources and, more recently, organic residues. Organic residue analysis identifies bee products by comparing the archaeological lipid residues to the lipid profile of modern beeswax. The lipid signature of fresh beeswax is characterized by its wax ester, saturated fatty acid, $n$-alkane and $n$-alcohol profiles. The long-chain even-numbered palmitate wax esters ($C_{40}$–$C_{52}$) are the most characteristic marker, maximizing at $C_{46}$ [13]. Saturated fatty acids are even-numbered and long-chain ($C_{14:0}$–$C_{36:0}$) with a predominance of lignoceric acid ($C_{24:0}$). Once degraded, there is an increasing amount of palmitic acid ($C_{16:0}$) formed from the hydrolysis of palmitate wax esters [13–15]. Beeswax is also characterized by odd-numbered $n$-alkanes ranging from $C_{21}$–$C_{35}$, maximizing at $C_{27}$ [13,15], as well as even-numbered $n$-alcohols ranging from $C_{24}$–$C_{36}$, maximizing at $C_{30}$ [15].

Using this biomolecular approach, beeswax was widely identified in the global archaeological record. Evidence for the exploitation of beeswax was found already in Middle Palaeolithic Europe and South Africa, where it was used for hafting flint tools by Neanderthals and anatomically modern humans [16–18]. A wider exploitation was documented during the 7th–3rd millennium cal. BC. Beeswax was used in dentistry [5] and hafting [19], and while no specific function was proposed, beeswax lipid residues were found in relation to pottery at over 50 Neolithic sites in Eurasia and North Africa (figure 1a; [20] and see references therein). The frequency of beeswax residue at these sites is highly variable, ranging from 1% to well over 30% of the tested vessels with significant lipid yield [20] (although the proportion may be influenced and biased by the number and types of vessels sampled as well as the preservation conditions offered by the climate and sediment characteristic of each site). The tested sites are from topographically and climatically variable areas, showing that bees thrived in diverse environmental and ecological niches and that their exploitation was a common and shared practice.

With a few potential south Levantine Late Chalcolithic (ca 4500–3700 cal. BC) exceptions [38], indications for the use of beeswax before the 3rd millennium BC in the wider Levantine region are rare. The first evidence for bees stems from later written sources. Textual evidence from the Egyptian First Dynasty described the Egyptian god Min as the 'master of wild bees' [1] and the king of Upper and Lower Egypt as 'he of the sedge and bee' [39]. Shortly thereafter (ca 2400 BC), beekeeping was depicted in bas-reliefs [1]. In Anatolia and Mesopotamia, bees were also highly valued. Evidence from Hittite legal documents described the fine for beehive theft as lofty and nearly equal to the fine for stealing a sheep [1,40], and other texts suggest that Sumerian honey was used as religious offerings [1]. Honey was also mentioned numerous times in the Bible, which described Israel as 'a land flowing with milk and honey' [Exodus 3.8]. Archaeological evidence for beekeeping in the southern Levant was first noted at the Iron Age II site of Tel Rehov (tenth–ninth century BC), located in the Jordan Valley, Israel. The site includes several beehives formed from hollow clay cylinders that would have been capable of producing hundreds of kilograms of honey annually. Their use as hives was reinforced by organic residue analysis (ORA) [41].

Other contemporary and slightly earlier examples of beeswax in the southern Levant were recently identified based on lipids preserved within ceramics and characterized by Gas Chromatography-Mass Spectrometry (GC-MS). Some of these identifications relied primarily or exclusively on the $n$-alkane profiles, and as such, beeswax identification is tenuous. Highly degraded beeswax was suggested as identified on a Late Bronze Age II clay coffin from Tel Shadud in the Jezreel Valley, Israel. Beeswax may have been applied to the coffin as a coating or a sealant [42]. Similar markers were observed from a storage jar found at the Late Bronze Age III site of Tel Azekah in the Shephelah, and it was suggested that these markers were related to bee products and that the jar was used to store honey [43]. Potential evidence for beeswax was also recovered from the Iron Age I burial site of Horvat Tevet in the Jezreel Valley, and it was suggested that heated beeswax was used in burial ceremonies [44]. Finally, a sub-set of beeswax markers was recovered from a large storage jar found at Jneneh, an Iron Age II site in north-central Jordan. The ceramic form is commonly associated with liquid storage, so it is possible that beeswax was used as a sealant; alternatively, the jar could have been used to store honey [45].

Beeswax is rare in the organic record of the prehistoric southern Levant although prior studies analysed ceramic vessels from many sites (figure 1b); beeswax residue has only potentially been identified in relation to cornets—a cone-shaped vessel characteristic of the Late Chalcolithic period. Cornets with lipid profiles similar to beeswax residue were recovered from three sites: Grar in the northern Negev and Moringa Cave and Ein Gedi in the Judean Desert, west of the Dead Sea. Based on the $n$-alkane profile, it was suggested that the cornets represent lamps used with beeswax candles

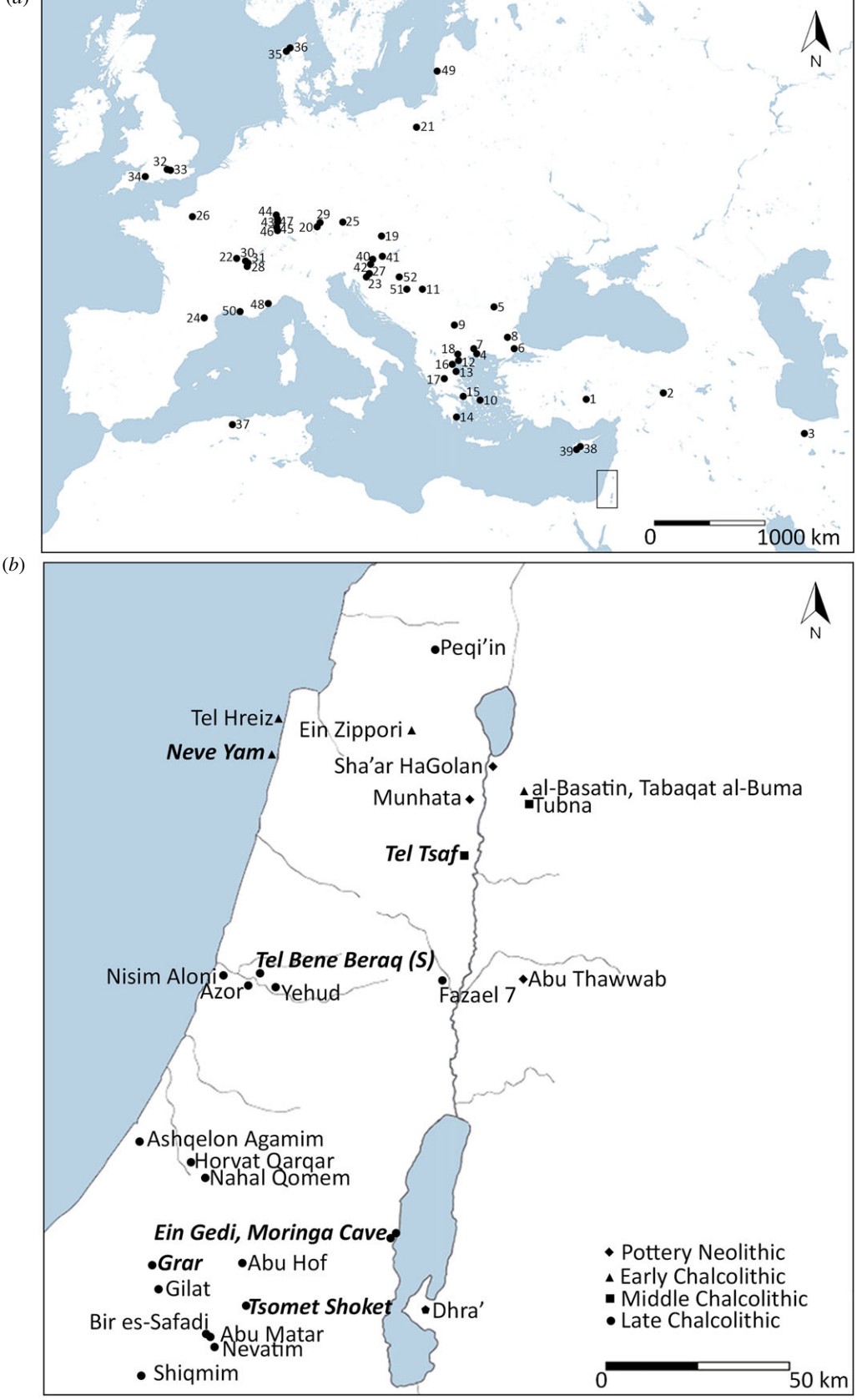

**Figure 1.** (*Caption overleaf..*)

**Figure 1.** (*Overleaf.*) (*a*) Map of sites from the 7th–3rd millennium BC with lipid biomarker evidence for beeswax: 1. Çatalhöyük [20]; 2. Çayönü Tepesi [20]; 3. Tepe Sofalin [21]; 4. Liménaria [22]; 5. Măgura [20]; 6. Toptepe [20]; 7. Dikili Tash [20,23]; 8. Aşağı Pınar [20]; 9. Drenovac Turska Cesma [20]; 10. Ftélia [20]; 11. Vinča Belo Brdo [20]; 12. Paliambela [20]; 13. Rachmani [20]; 14. Kouvéléikès A [20]; 15. Profitis Ilias Rizoupolis [20]; 16. Vassilara Rachi [20]; 17. Théopetra [20]; 18. Balkan Export [20]; 19. Brunn am Gebirge [20]; 20. Niederhummel [20]; 21. Kuyavia region [24]; 22. Chassey-le-Camp [20]; 23. Moverna vas [20]; 24. Font-Juvénal [20]; 25. Künzing-Unternberg [20]; 26. Bercy [14]; 27. Ajdovska jama [20,25]; 28. Clairvaux XIV [20]; 29. Ergolding Fischergasse [26]; 30. Chalain 3 [20,27,28]; 31. Chalain 4 [20,27,28]; 32. Eton rowing lake [29]; 33. Runnymede bridge [20]; 34. Bulford Torstone [20]; 35. Åle [20]; 36. Bjørnsholm [20]; 37. Gueldaman [20]; 38. Sotira Teppes [30]; 39. Erimi Pamboula [30]; 40. Zgornje Radvanje [31]; 41. Turnišče [31]; 42. Spodnje Hoče [31]; 43. Bischoffsheim [32]; 44. Rosheim [32]; 45. Ensisheim [32]; 46. Sierentz [32]; 47. Colmar [32]; 48. Pendimoun rock-shelter [33]; 49. Šventoji 4 [34]; 50. Vernègues-Cazan-l'Heritière [35]; 51. Starčevo-Grad [36]; 52. Magareći Mlin [36]; (*b*) Map of sites in the southern Levant dating to ca. 6400–3700 BC [37] whose pottery assemblages were studied using ORA. Sites with evidence for beeswax are in bold and italics: Neve Yam (current study), Tel Tsaf (current study) Tel Bene Beraq (South) (current study), Moringa Cave [38], Ein Gedi [38], Grar [38] and Tsomet Shoket (current study).

[38], although the *n*-alkane profile alone is insufficient to support this with certainty. These few examples do not sufficiently explain the emergence of intensive bee exploitation in the southern Levant observed in the Late Bronze Age and Iron Age. Further archaeological and palynological evidence (e.g. [46]) to suggest the presence of beehives or bees is also absent, but additional archaeological investigation and palynological analysis is required.

The aim of this study is to use a lipid biomarker approach to investigate a possible earlier presence and use of beeswax in the Chalcolithic period of the southern Levant. This research is part of a wider study, which tested 247 ceramic vessels from 15 prehistoric sites in Israel using ORA. Emphasis was placed on pottery from the Chalcolithic period (including the Early Chalcolithic (*ca* 5800–5300 cal. BC), Middle Chalcolithic (*ca* 5300–4700 cal. BC) and Late Chalcolithic (*ca* 4500–3700 cal. BC)), with a few vessels from the Pottery Neolithic (*ca* 6400–5800 cal. BC) also tested. At the end of this timespan, important socio-economic transformations and technological advances occurred, such as developments in olive horticulture, the intensified use of secondary products, craft specialization and the introduction of copper metallurgy (e.g. [47–51]). The results obtained here together with previously published research are discussed to explain why during this timespan beehive products were used and, more importantly, why this versatile resource appears to have been underexploited.

## 2. Material and methods

In the course of the wider study, ceramic vessels were sampled from a geographically and chronologically diverse range of sites (table 1). Vessels were selected from ongoing excavations and assemblages stored in the Israel Antiquities Authority's collection. There was no targeted selection of specific vessels as prior studies on south Levantine sites identified beeswax inconsistently in both task-specific and unspecialized vessel forms, such as bowls, jars, cornets, chalices, flasks and lamps [38,43,44]. Correspondingly, we sampled a wide selection of vessel forms (e.g. bowls, holemouth jars, necked jars, bow-rim jars, pithos spouted vessels, platters, churns, cornets and strainers), which are commonly found at prehistoric sites [52], as well as typologically unidentified fragments.

Following cleaning the initial surface layer (*ca* 1–2 mm) to remove potential exogenous contamination, around 1–2 g of ceramic powder were collected from the internal surface of each vessel using a Dremel modelling drill fitted with a tungsten bit. An additional crust from an internal vessel surface was collected and analysed after being homogenized. Ceramic powder from the external surfaces or soil samples from the layer the ceramics were retrieved from were collected (*ca* 1–2 g) and analysed as controls to test for exogenous and post-depositional contamination.

Glassware was sterilized before use (500°C in a muffle oven overnight), and HPLC grade solvents were used. Lipids were extracted alongside a process blank to monitor for laboratory contamination. The 'unbound' lipid fraction was recovered by solvent extraction (following [53]), and this was followed for internal samples by saponification to release the 'bound' lipid fraction (following a modified Method D in [54]). This two-step approach was selected in favour of a direct acidified methanol extraction [55] to ensure that wax esters and acylglycerols could be analysed before hydrolysis.

Lipids were extracted three times using 5 ml of dichloromethane : methanol (DCM:MeOH, 2:1, *v:v*), assisted by sonication (15 min) and centrifugation (10 min at 3000 r.p.m.). The solvent was dried under a gentle stream of nitrogen and mild heating (40°C) to obtain the total lipid extract (TLE). An aliquot of

**Table 1.** Summary of the sites analysed by ORA.

| site | period | N pottery vessels sampled | types of vessels sampled |
| --- | --- | --- | --- |
| Sha'ar HaGolan | Pottery Neolithic | 16 | bowls, jars and typologically unidentified vessel fragments |
| Neve Yam | Early Chalcolithic | 17 | jars, bow-rim jars, spouted vessels and typologically unidentified vessel fragments |
| Tel Hreiz | Early Chalcolithic | 3 | jars and typologically unidentified vessel fragments |
| Tel Tsaf | Middle Chalcolithic | 100 | bowl, v-shaped bowls, deep bowls, small bowls, hemispherical bowls, jars, holemouth jars, necked jars, platters and typologically unidentified vessel fragments |
| Abu Matar | Late Chalcolithic | 3 | holemouth jars and churns |
| Ashqelon Agamim | Late Chalcolithic | 28 | bowls, jars, cornets and typologically unidentified vessel fragments |
| Azor | Late Chalcolithic | 1 | strainer vessel |
| Tel Bene Beraq (South) | Late Chalcolithic | 10 | jars, holemouth jars and typologically unidentified vessel fragments |
| Bir es-Safadi | Late Chalcolithic | 11 | jars, holemouth jars, spouted vessels and churns |
| Fazael 7 | Late Chalcolithic | 8 | jars, holemouth jars and typologically unidentified vessel fragments |
| Gilat | Late Chalcolithic | 13 | v-shaped bowls, jars, holemouth jars, churns and cornets |
| Nisim Aloni | Late Chalcolithic | 9 | bowls, v-shaped bowls, jars, churns and typologically unidentified vessel fragments |
| Peqi'in | Late Chalcolithic | 3 | v-shaped bowls |
| Tsomet Shoket | Late Chalcolithic | 18 | bowls, v-shaped bowls, jars, holemouth jars, pithoi and churns |
| Yehud | Late Chalcolithic | 7 | churns, strainer vessels and typologically unidentified vessel fragments |

50% of the TLE was silylated using 50 µl of N,O-bis (trimethylsilyl) trifluoroacetamide with 1% trimethylchlorosilane and 4 µl of pyridine (40°C; 30 min). The samples were run as trimethylsilylated (TMS) derivatives. After derivitization, initially a known amount of hexatriacontane ($C_{36}$ n-alkane; 98%, Sigma-Aldrich) was added to the samples to allow for TLE quantification; upon noticing that hexatriacontane occasionally co-eluted with other molecules, the methodology was changed, and it was run systematically in the GC-MS sequence after every four samples to allow for a more accurate quantification.

Following solvent extraction, the lipids bound to the ceramic matrix were saponified (70°C; 90 min). 4 ml of 0.5 M sodium hydroxide solution ($MeOH:H_2O$, 9.1, v:v) was used. The neutral fraction was extracted three times with cyclohexane and was not analysed further. To the remaining sodium hydroxide solution, 2 ml of 1 M hydrochloric acid was added until pH 3 was reached. The acid fraction was extracted three times with cyclohexane. It was then methylated using 250 µl of boron trifluoride methanolic solution (14%) (70°C; 60 min). The samples were run as methyl esters. As with the solvent extracts, initially a known amount of $C_{36}$ n-alkane was added to the samples to allow for TLE quantification, and in later batches, the $C_{36}$ n-alkane was run only externally after every four archaeological samples.

GC-MS was performed using an Agilent 7890B GC coupled to an Agilent 5977A Mass Spectrometer (MSD) and Flame Ionization Detector (FID). Injections were carried out using a GERSTEL multi-purpose sampler and a GERSTEL Cold-Injection System (CIS) 4. The samples were run on an Agilent J&W DB-5HT column (15 m × 0.32 mm i.d.; 0.1 µm film thickness), and the eluent was divided into two equal parts using 0.18 mm non-coated deactivated silica capillary columns (0.66 m splitter-column to the FID and 1.52 m splitter-column to the MSD) with the Three-Way Splitter Kit. The inlet temperature was ramped from 30°C to 240°C at 12°C s$^{-1}$ (held isothermally for 5 min) and then increased to 350°C at 12°C s$^{-1}$ (held isothermally for 10 min). The oven temperature was ramped from 40°C (held isothermally for 1 min) to 100°C at 15°C min$^{-1}$ and then to 240°C at 6°C min$^{-1}$ and then to 350°C at 10°C min$^{-1}$ (held isothermally for 20 min). The analysis employed helium as the carrier gas. Splitless injection was applied, with a purge flow of 3.0 ml min$^{-1}$ and a constant pressure at the head of the column of 8.4435 psi. Mass spectra were acquired using electron ionization at 70 eV. The mass range acquired was from m/z 50–950 in 1.562 s. The temperatures of the ion source, transfer line and FID were each 300°C. Mass spectra were identified using the National Institute of Standards and Technology Library, 2014 edition, and n-alkanes were identified through comparison to a saturated n-alkane standard mix ($C_7$–$C_{40}$) (Sigma-Aldrich).

# 3. Results

Lipid preservation was low as expected given the Mediterranean climate and alkaline soil [56] characteristic of the southern Levant. Both factors support microbial activity and correspondingly lipid degradation [57,58]. From the 247 vessels tested, only 22 vessel interiors and one calcified crust yielded more than 5 µg g$^{-1}$ of lipids after solvent extraction, considered an interpretable residue [59]. While originally sampled as controls, an additional seven vessels had more than 5 µg g$^{-1}$ of lipids exclusively externally. This may relate to spillage or more likely the application of a vessel treatment to alter the vessel surface or porosity [60] (post-depositional contamination, spillage and lipid migration from the vessel interior may likely be ruled-out because of the lack of lipids internally). The bound acid fraction had a much higher lipid yield. Within this fraction, 47% of the internal layers of the vessels tested (n = 116) and the crust contained over 5 µg g$^{-1}$ of lipid.

Among these internally and externally solvent extracted vessels, evidence for beeswax was identified in only four vessels from four of the studied sites (figure 2): Early Chalcolithic Neve Yam, Middle Chalcolithic Tel Tsaf and Late Chalcolithic Tel Bene Beraq (South) and Tsomet Shoket (see site details in electronic supplementary material, S1). This includes the interior surface of a jar from Neve Yam (NY-12 interior layer 2), the calcified crust on a wall fragment from Tel Tsaf (TSF18-57 crust), the interior surface of a jar from Tel Bene Beraq (South) (BB-21 interior layer 2) and the exterior surface of a jar from Tsomet Shoket (TS-4 exterior) (table 2). The related biomarker evidence, which will be detailed below, includes even-numbered long-chain saturated fatty acids, odd-numbered long-chain n-alkanes, even-numbered long-chain n-alcohols and palmitate wax esters [13,15].

Beeswax markers were not present in a variety of other vessel forms tested, including bowls, holemouth jars, bow-rim jars, spouted vessels, platters, churns, cornets and strainers. The absence of beeswax markers in the six cornets analysed is notable considering the proposed positive

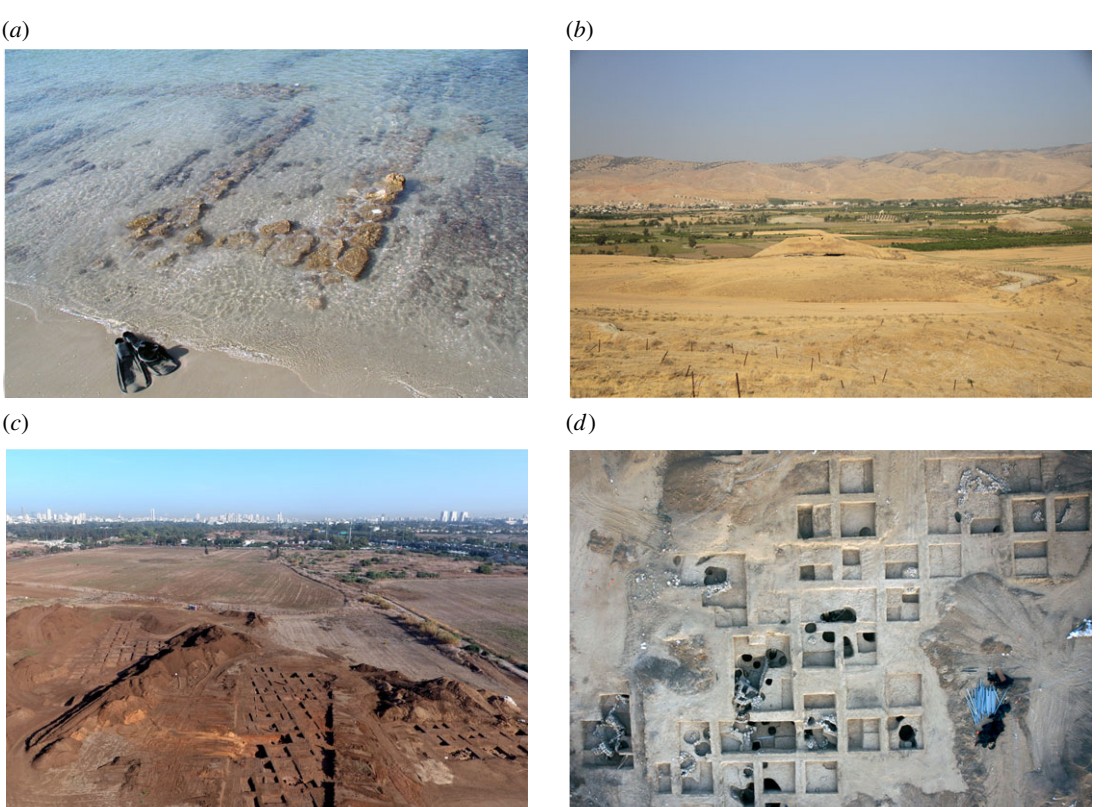

**Figure 2.** A view of the sites with evidence for beeswax: (*a*) Neve Yam (photo by E. Galili); (*b*) Tel Tsaf (photo by D. Rosenberg); (*c*) Tel Bene Beraq (South) (photo by I. Marmelstein (Israel Antiquities Authority)); (*d*) Tsomet Shoket (photo by A. Peretz (Israel Antiquities Authority)).

identification in an earlier study based on the *n*-alkane profiles [38]. This suggests that if cornets were used as candles, other fuel sources were used in addition to beeswax. Alternatively, the beeswax biomarkers in the cornets could relate to an entirely different use of beeswax, for instance as a sealant for drinking vessels (e.g. [61]).

A more in-depth biomarker analysis of the four vessels with evidence for beeswax was conducted to confirm this attribution and identify additional lipid inputs:

## 3.1. Jar wall fragment from Neve Yam (NY-12)

The TLE of the interior surface of the jar found at Neve Yam (NY-12 interior layer 2; figure 3; table 2) contained in part saturated fatty acids, including pelargonic ($C_{9:0}$), pentadecylic ($C_{15:0}$), palmitic ($C_{16:0}$), margaric ($C_{17:0}$) and stearic ($C_{18:0}$) acid, maximizing at $C_{16:0}$. The bound acid fraction revealed a wider range of saturated fatty acids ($C_{12:0}$–$C_{30:0}$), with high amounts of long-chain even-numbered fatty acids (electronic supplementary material S2). The *n*-alkanes in the TLE ranged from $C_{20}$–$C_{29}$, maximizing at *n*-nonacosane ($C_{29}$), with a clear odd over even preference as identified by the carbon preference index (CPI) (CPI 28.1) (following [62]). The *n*-alcohols ranged from $C_{12}$–$C_{30}$; all are even, and they maximize at 1-octacosanol ($C_{28}$). Even-numbered palmitate wax esters with 42–46 carbon atoms were present, with $C_{44}$ the most abundant.

Some of the lipid compounds indicate contributions from animal fat and plant oil. The presence of animal fat may be indicated by the recovery of cholesterol [63]. No cholesterol degradation markers were identified, so this may suggest contamination, although no other contaminants that originate from human contact (e.g. [64]) were identified. Ruminant fat may be specifically suggested based on the odd-numbered and branched-chain fatty acids in the acid fraction as well as the high amount of $C_{18:0}$ (electronic supplementary material, S2; table 2) [65,66]. Odd-numbered and branched-chain fatty acids are formed by bacteria in the rumen [66]. The presence of a plant oil is suggested by the unsaturated fatty acids in the solvent extract ($C_{18:1}$, two isomers) and acid fraction (one isomer of $C_{16:1}$

**Table 2.** Lipid profile summary of the samples with evidence for beeswax and their controls (SE, solvent extract; AF, acid fraction; K, ketone). The maximum molecule in each category is bolded and the lipid marker sets that are characteristic of fresh beeswax are underlined.

| vessel | sample | TLE (µg g⁻¹) SE | TLE (µg g⁻¹) AF | saturated fatty acids SE | saturated fatty acids AF | unsaturated fatty acids SE | unsaturated fatty acids AF | branched-chain fatty acids in AF | N-alkanes in SE | N-alcohols in SE | wax esters in SE | other molecules in SE and AF |
|---|---|---|---|---|---|---|---|---|---|---|---|---|
| **NY-12** | **interior layer 2** | 5.0 | 58.9 | $C_{9:0}$ $C_{15:0}$ **$C_{16:0}$, $C_{17:0}$** <u>$C_{18:0}$</u> | $C_{12:0}$ $C_{13:0}$ $C_{14:0}$ $C_{15:0}$ <u>$C_{16:0}$</u> <u>**$C_{18:0}$**</u> $C_{19:0}$ $C_{20:0}$ $C_{21:0}$ $C_{22:0}$ $C_{23:0}$ $C_{24:0}$ $C_{25:0}$ $C_{26:0}$ $C_{27:0}$ $C_{28:0}$ $C_{30:0}$ | $C_{18:1}$ | $C_{16:1}$ $C_{18:1}$ $C_{20:1}$ | $C_{14:0}$ $C_{16:0}$ | $C_{20}$ $C_{21}$ $C_{23}$ $C_{24}$ $C_{25}$ <u>$C_{27}$</u> <u>**$C_{29}$**</u> | $C_{12}$ $C_{14}$ $C_{16}$ $C_{18}$ $C_{24}$ $C_{26}$ <u>**$C_{28}$**</u> $C_{30}$ | <u>$C_{42}$</u>, <u>**$C_{44}$**</u> | K₃₁, K₃₃, 1-monopalmitin, Cholesterol |
| | **exterior** | 1.4 | — | <u>**$C_{16:0}$**</u>, <u>$C_{18:0}$</u> | — | — | — | — | $C_{20}$ $C_{21}$ $C_{22}$ $C_{23}$ $C_{24}$ $C_{25}$ $C_{26}$ $C_{27}$ $C_{28}$ <u>**$C_{29}$**</u> $C_{31}$ | $C_{14}$ $C_{16}$ $C_{18}$ $C_{20}$ $C_{22}$ $C_{24}$ <u>**$C_{26}$**</u> $C_{28}$ | <u>$C_{42}$</u>, <u>**$C_{44}$**</u> | 1-monopalmitin |
| **TSF18-57** | **TSF18-57 internal crust** | 6.8 | 12.8 | <u>$C_{16:0}$</u>, <u>$C_{18:0}$</u> | <u>$C_{14:0}$</u> <u>$C_{15:0}$</u> <u>$C_{16:0}$</u> <u>$C_{17:0}$</u> **$C_{18:0}$**, $C_{19:0}$ $C_{20:0}$ $C_{21:0}$ $C_{22:0}$ $C_{23:0}$ $C_{24:0}$ $C_{25:0}$ $C_{26:0}$ $C_{27:0}$ $C_{28:0}$ $C_{30:0}$ | — | $C_{18:1}$ $C_{18:2}$ | $C_{14:0}$–$C_{16:0}$ | <u>$C_{23}$</u> <u>$C_{25}$</u> <u>$C_{26}$</u> <u>$C_{27}$</u> $C_{28}$ **$C_{29}$** $C_{30}$ $C_{31}$ $C_{32}$ **$C_{33}$** | $C_{20}$ $C_{21}$ $C_{22}$ $C_{23}$ $C_{24}$ $C_{25}$ $C_{26}$ $C_{27}$ **$C_{28}$** $C_{30}$ $C_{32}$ $C_{34}$ | $C_{40}$, $C_{42}$ <u>**$C_{44}$**</u>, <u>$C_{46}$</u> $C_{48}$ | K₃₁ |
| | **TSF18-57 interior layer 2** | 3.1 | 5.6 | $C_{9:0}$ $C_{10:0}$ $C_{15:0}$ **$C_{16:0}$**, $C_{18:0}$ $C_{20:0}$ $C_{22:0}$ $C_{24:0}$ $C_{26:0}$ $C_{28:0}$ | $C_{14:0}$ $C_{15:0}$ $C_{16:0}$ $C_{17:0}$ **$C_{18:0}$**, $C_{19:0}$ $C_{20:0}$ $C_{21:0}$ $C_{22:0}$ $C_{23:0}$ $C_{24:0}$ $C_{25:0}$ $C_{26:0}$ $C_{27:0}$ $C_{28:0}$ | $C_{18:1}$ | $C_{18:1}$ $C_{18:2}$ | - | <u>$C_{23}$</u> <u>$C_{25}$</u> <u>**$C_{27}$**</u> <u>$C_{29}$</u> $C_{31}$ | $C_{18}$ $C_{20}$ $C_{22}$ $C_{23}$ $C_{24}$ $C_{26}$ $C_{27}$ **$C_{28}$** $C_{30}$ $C_{32}$ | $C_{40}$, $C_{42}$ <u>**$C_{44}$**</u>, <u>$C_{46}$</u> | K₃₁, 1-monopalmitin, 1-monostearin |
| | **TSF18-57 exterior** | 1.5 | — | <u>$C_{16:0}$</u>, <u>**$C_{18:0}$**</u> <u>$C_{28:0}$</u> | — | — | — | — | $C_{14}$ $C_{23}$ $C_{25}$ $C_{26}$ $C_{27}$ **$C_{29}$** $C_{30}$ $C_{31}$ $C_{33}$ | $C_{22}$ $C_{24}$ $C_{26}$ $C_{27}$ **$C_{28}$** $C_{30}$ $C_{32}$ | — | — |

**Table 2.** (*Continued.*)

| vessel | sample | TLE (μg g⁻¹) | | saturated fatty acids | | unsaturated fatty acids | | branched-chain fatty acids in AF | N-alkanes in SE | N-alcohols in SE | wax esters in SE | other molecules in SE and AF |
|---|---|---|---|---|---|---|---|---|---|---|---|---|
| | | SE | AF | SE | AF | SE | AF | | | | | |
| **BB-21** | **interior layer 2** | 55.3 | 86.4 | $C_{16:0}$, $C_{18:0}$, $C_{24:0}$ | $C_{9:0}$–$C_{13:0}$, $C_{14:0}$, $C_{15:0}$, $C_{16:0}$, $C_{18:0}$, $C_{20:0}$, $C_{21:0}$, $C_{22:0}$, $C_{23:0}$, $C_{24:0}$, $C_{25:0}$, $C_{26:0}$, $C_{27:0}$, $C_{28:0}$, $C_{30:0}$ | — | $C_{18:1}$ | $C_{14:0}$, $C_{17:0}$ | $C_{23}$, $C_{24}$, $C_{25}$, $C_{26}$, $C_{27}$, $C_{28}$, $C_{29}$, $C_{33}$, $C_{35}$, $C_{37}$ | $C_{24}$, $C_{26}$, $C_{28}$, $C_{30}$, $C_{32}$ | $C_{40}$, $C_{42}$, $C_{44}$, $C_{46}$, $C_{48}$, $C_{50}$ | $K_{33}$, $C_{14}$–$C_{30}$ dicarboxylic acids |
| | **BB-21 soil** | 0.7 | — | $C_{16:0}$, $C_{18:0}$, $C_{22:0}$, $C_{24:0}$, $C_{26:0}$ | — | — | — | — | $C_{27}$, $C_{29}$ | $C_{16}$, $C_{18}$, $C_{22}$, $C_{24}$, $C_{26}$, $C_{28}$, $C_{30}$ | — | 1-monopalmitin, β-Sitosterol, Campesterol, Stigmasterol, Ferruginol, 2,3-Dehydroferruginol, Sempervirol, Totarol |
| **TS-4** | **interior layer 2** | 0.6 | 2.8 | $C_{16:0}$ | $C_{14:0}$, $C_{15:0}$, $C_{16:0}$, $C_{17:0}$, $C_{18:0}$, $C_{20:0}$, $C_{22:0}$, $C_{23:0}$, $C_{24:0}$ | — | $C_{16:1}$, $C_{18:1}$, $C_{18:2}$, $C_{20:1}$, $C_{22:1}$ | $C_{14:0}$, $C_{15:0}$ | $C_{27}$, $C_{29}$ | $C_{16}$, $C_{18}$ | — | 1-monopalmitin, β-Sitosterol, Cholesterol |
| | **TS-4 exterior** | 6.6 | — | — | — | — | — | — | $C_{25}$, $C_{26}$, $C_{27}$, $C_{28}$, $C_{29}$, $C_{30}$, $C_{31}$, $C_{33}$ | $C_{18}$, $C_{24}$, $C_{26}$, $C_{28}$, $C_{30}$, $C_{32}$ | $C_{40}$, $C_{42}$, $C_{44}$, $C_{46}$ | — |

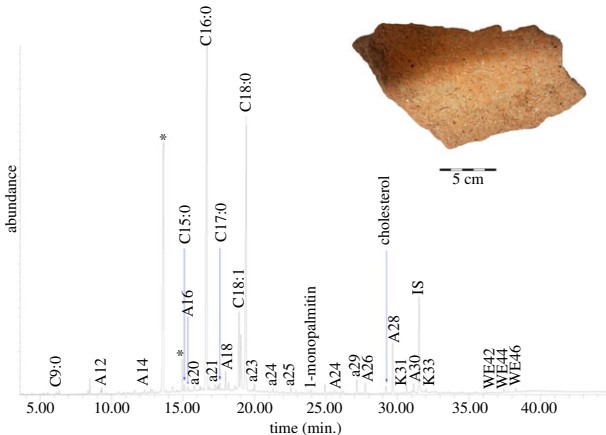

**Figure 3.** Total Ion Chromatogram (TIC) of the solvent extracted TLE released from the interior surface of a jar wall fragment from Neve Yam (NY-12 interior layer 2) analysed as TMS derivatives (*, plasticisers; ax, $n$-alkane with chain length X; Ax, $n$-alcohol with chain length X; Cx:y, fatty acid with chain length X and Y number of double bonds; Kx, ketone with chain length X; WEx, wax ester with chain length X).

and $C_{20:1}$ and two isomers of $C_{18:1}$), although small amounts of $C_{18:1}$ are also present in animal fat [67,68]. Ketones were also identified, including hentriacontan-16-one ($K_{31}$) and tritriacontan-16-one ($K_{33}$).

The external control sample (NY-12 exterior) had many of the same molecules in insignificant amounts, suggesting that lipids may have migrated through the ceramic matrix and that the lipid signature in the interior did not result from post-depositional contamination. This includes $C_{16:0}$, $C_{18:0}$, $n$-alkanes ($C_{20}$–$C_{31}$), even-numbered $n$-alcohols ($C_{14}$–$C_{28}$) and palmitate wax esters ($C_{42}$ and $C_{44}$) (electronic supplementary material, S2; table 2).

## 3.2. Crust on wall fragment from Tel Tsaf (TSF18-57)

The TLE of a crust from Tel Tsaf (TSF18-57 crust; figure 4; table 2) contains trace amounts of saturated fatty acids, including $C_{16:0}$ and $C_{18:0}$. In the bound acid fraction, significant amounts of saturated fatty acids were identified, ranging from $C_{14:0}$–$C_{30:0}$, with high amounts of long-chain even saturated fatty acids (electronic supplementary material, S2). In the TLE, the $n$-alkanes ranged from $C_{23}$–$C_{33}$, with $C_{29}$ the most abundant. There is a clear odd over even preference (CPI 9.4). The $n$-alcohols ranged from $C_{20}$–$C_{34}$ and maximize at $C_{28}$. The few odd-numbered $n$-alcohols present were identified in trace amounts. Palmitate wax esters are even-numbered with 40–48 carbons atoms. $C_{44}$ is the most abundant wax ester.

Additional markers in the solvent extraction and bound acid fraction are suggestive of animal fat and plant oil. Ruminant fat may be suggested based on the acid fraction, which contains high amounts of $C_{18:0}$, odd-numbered saturated fatty acids and branched-chain fatty acids (electronic supplementary material, S2; table 2) [65,66]. $C_{18:1}$ (two isomers) and $C_{18:2}$ (one isomer) were also identified and may be used to suggest a potential plant oil (e.g. [68]), although $C_{18:1}$ may also originate from animal fat [67,68]. Additionally, a single ketone was identified – $K_{31}$.

The TLE of the internal ceramic sample (TSF18-57 interior layer 2; electronic supplementary material, S2; table 2) and the external control sample (TSF18-57 exterior; electronic supplementary material 2; table 2) had an insignificant lipid yield. In the TLE of the internal ceramic sample, there is a wide range of saturated fatty acids with 9–28 carbon atoms; many of these same fatty acids were released by saponification, paralleling those in the crust. The identified odd-numbered $n$-alkanes ($C_{23}$–$C_{31}$), even-numbered $n$-alcohols ($C_{18}$–$C_{32}$), even-numbered palmitate wax esters ($C_{40}$–$C_{46}$) and $K_{31}$ also parallel the molecules in the crust. This suggests that the lipids were absorbed into the ceramic matrix during the use episode that formed the crust. In the external sample, there is $C_{16:0}$, $C_{18:0}$, $C_{28:0}$, a wide range of $n$-alkanes, including $C_{14}$–$C_{33}$, and $n$-alcohols, with $C_{22}$–$C_{32}$. Wax esters were not identified on the vessel exterior, reinforcing that these originated from vessel use and not post-depositional contamination.

## 3.3. Jar wall fragment from Tel Bene Beraq (South) (BB-21)

The TLE from the interior surface of the jar found at Tel Bene Beraq (South) (BB-21 interior layer 2; figure 5; table 2) contains the following saturated fatty acids: $C_{16:0}$, $C_{18:0}$ and $C_{24:0}$. In the bound

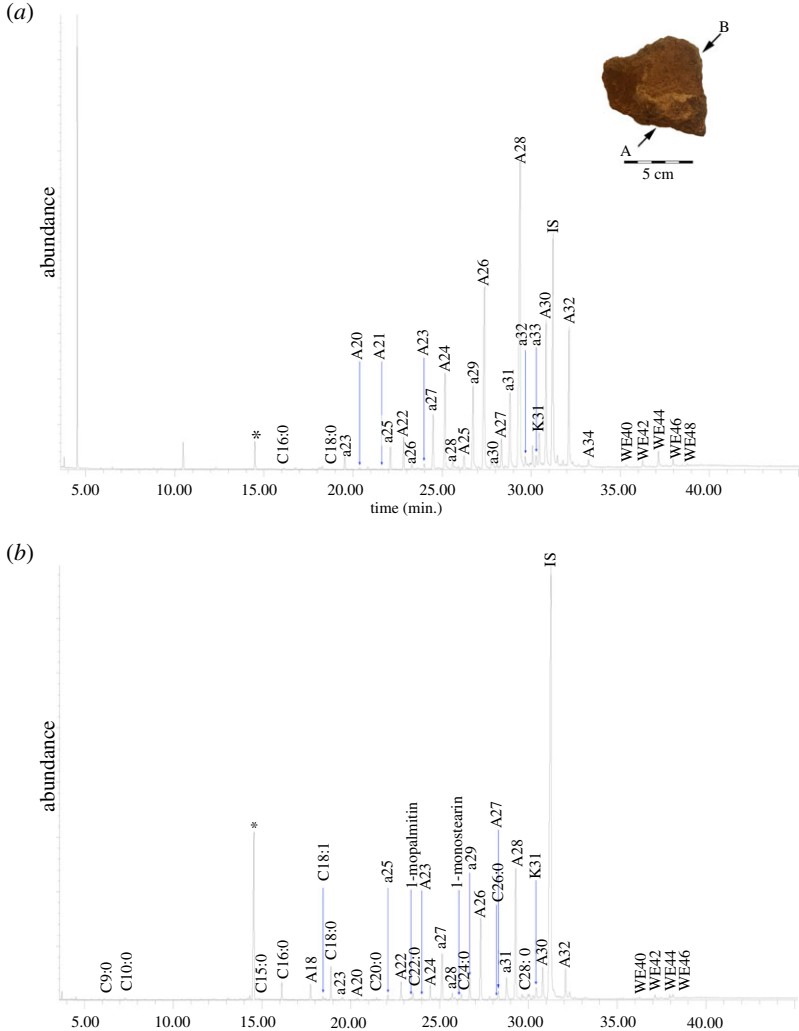

**Figure 4.** TIC of the solvent extracted TLE released from (*a*) the crust (TSF18-57 internal crust); (*b*) the interior wall (TSF18-57 interior layer 2) of a fragment from Tel Tsaf analysed as TMS derivatives (*, plasticisers; a*x*, *n*-alkane with chain length X; A*x*, *n*-alcohol with chain length X; C*x:y*, fatty acid with chain length X and Y number of double bonds; K*x*, ketone with chain length X; WE*x*, wax ester with chain length X).

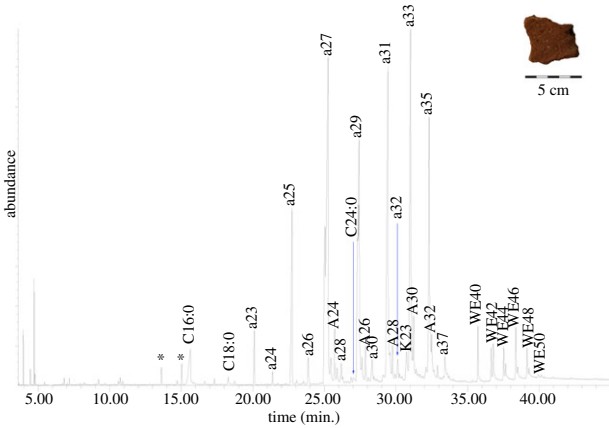

**Figure 5.** TIC of the solvent extracted TLE released from the interior surface of a jar wall fragment from Tel Bene Beraq (South) (BB-21 interior layer 2) analysed as TMS derivatives (*, plasticisers; a*x*, *n*-alkane with chain length X; A*x*, *n*-alcohol with chain length X; C*x:y*, fatty acid with chain length X and Y number of double bonds; K*x*, ketone with chain length X; WE*x*, wax ester with chain length X).

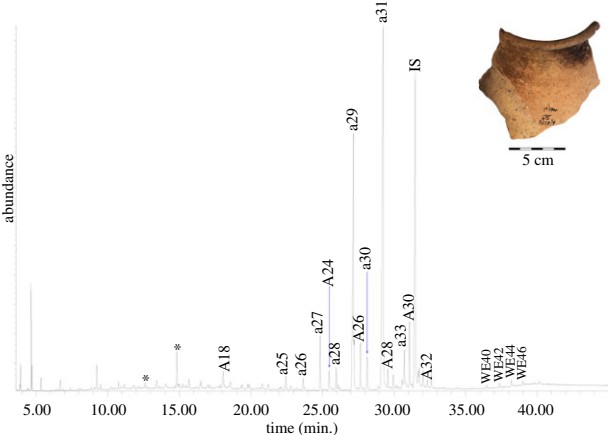

**Figure 6.** TIC of the solvent extracted TLE released from the exterior surface of a jar rim fragment from Tsomet Shoket (TS-4 exterior) analysed as TMS derivatives (*, plasticisers; ax, $n$-alkane with chain length X; Ax, $n$-alcohol with chain length X; IS, internal standard; WEx, wax ester with chain length X).

fraction, there was a wider range of saturated fatty acids with 9–28 carbon atoms (electronic supplementary material S2). Primarily, these are long-chain and even-numbered, and they maximize at $C_{24:0}$. The $n$-alkanes in the TLE ranged from $C_{23}$–$C_{37}$, maximizing at $n$-heptacosane ($C_{27}$) with a predominance of odd-numbered $n$-alkanes (CPI 23.3). The $n$-alcohols are exclusively even and range from $C_{24}$–$C_{32}$, maximizing at 1-triacontanol ($C_{30}$). Wax esters are even-numbered and palmitate, with 40–50 carbon atoms, maximizing at $C_{46}$. Most of the wax esters ($C_{42}$–$C_{50}$) have a variant eluting immediately after. Based on the retention time, these should be hydroxy wax esters (e.g. [26,69]), but the mass spectras lack a clear base peak of $m/z$ 117, deterring identification.

Additional markers link the sample to animal and plant products. In the acid fraction, high amounts of $C_{18:0}$, odd-numbered saturated fatty acids and branched-chain fatty acids are suggestive of ruminant adipose fat (electronic supplementary material, S2; table 2) [65,66]. $C_{18:1}$ and long-chain dicarboxylic acids with 14–30 carbon atoms were also identified in the acid fraction. Long-chain dicarboxylic acids are found naturally in different plant products (e.g. hardwood, nuts and some wild plants; [70–72]) and are unlikely oxidation products because very long-chain unsaturated fatty acids, which are rare, would be required to produce these. $K_{33}$ was also identified in the solvent extract.

By contrast, in a control soil sample from the same context (BB-21 soil; electronic supplementary material S2; table 2), there are trace amounts of lipids. These include fatty acids ($C_{16:0}$–$C_{26:0}$), $n$-alkanes ($C_{27}$ and $C_{29}$), $n$-alcohols ($C_{16}$–$C_{30}$), phytosterols and terpenoids (ferruginol, 2,3-dehydroferruginol, sempervirol and totarol). These are consistent with degraded plant products in the soil and do not match the archaeological signature. Specifically, the terpenoids may originate from certain conifers including *Cupressaceae* and *Cedrus* [73,74].

## 3.4. Jar rim fragment from Tsomet Shoket (TS-4)

In the jar from Tsomet Shoket, significant amounts of lipids were only identified in the TLE of the external surface (TS-4 exterior; figure 6; table 2), with traces of $C_{16:0}$, $n$-alkanes ($C_{27}$ and $C_{29}$) and $n$-alcohols ($C_{16}$ and $C_{18}$) in the interior sample (TS-4 interior layer 2; electronic supplementary material, S2; table 2). In the exterior, there are no fatty acids. $n$-Alkanes range from $C_{25}$–$C_{33}$, maximizing at triacontane ($C_{31}$). All are long-chain, and most are odd-numbered (CPI 16.5). $n$-Alcohols are even-numbered and range from $C_{18}$–$C_{32}$, maximizing at $C_{30}$. The palmitate wax esters are even with 40–46 carbon atoms, maximizing at $C_{44}$. Because the exterior was originally sampled as a control, it was not saponified, so it is unknown if fatty acids were bound to the ceramic matrix.

## 3.5. Comparison to modern beeswax

The identification of beeswax was reinforced by comparing the archaeological samples to fresh beeswax published in previous studies (figure 7) (following [13,15]). Fresh beeswax is characterized by an abundance of long-chain even-numbered fatty acids, long-chain odd-numbered $n$-alkanes, long-chain even-numbered $n$-alcohols and long-chain even-numbered palmitate wax esters [13,15]. As such, the

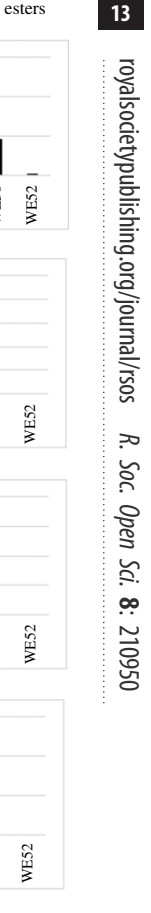

**Figure 7.** Histograms of the relative abundance of long-chain odd-numbered *n*-alkanes, even-numbered *n*-alcohols and palmitate wax esters of: (*a*) modern beeswax (following [13] tables 4, 6 and [15] table 2); and the archaeological samples with evidence for beeswax in this study: (*b*) Neve Yam; (*c*) Tel Tsaf; (*d*) Tel Bene Beraq (South) and (*e*) Tsomet Shoket.

fatty acid profile and the abundance of each odd-numbered *n*-alkane ($C_{21}$–$C_{35}$), even-numbered *n*-alcohol ($C_{24}$–$C_{36}$) and even-numbered palmitate wax ester ($C_{40}$–$C_{52}$) were calculated and compared. Parallels can be observed in all four samples, but the identification of beeswax for NY-12 and TS-4 is a bit more tenuous.

Comparison of the saturated fatty acid profiles to fresh beeswax [13–15] shows some clear parallels. In two of the solvent extracts and in all of the acid fractions, $C_{16:0}$ is abundant (although not always the maximum saturated fatty acid), reflecting hydrolysis of the palmitate wax esters. The long-chain even-numbered fatty acids characteristic of beeswax were primarily preserved bound to the ceramic matrix (electronic supplementary material S2; table 2), and in the unbound lipid fraction, they were unpreserved. In the TLE of the jar from Tsomet Shoket and the crust from Tel Tsaf, fatty acids are less common and may have been lost through sublimation [14].

Like fresh beeswax (figure 7*a*), in the archaeological samples, the *n*-alkanes are primarily odd-numbered and long-chain, maximizing at $C_{27}$, $C_{29}$ or $C_{31}$ (figure 7*b*–*e*). The shorter odd-numbered *n*-alkanes that are characteristic of beeswax were less common in the archaeological samples and were seemingly lost due to the decay processes. The longer-chain *n*-alkanes likely preferentially preserved due to their decreased volatility [14] and their greater abundance in fresh beeswax. In a few samples, there are short-chain *n*-alkanes that suggest bacterial input [75] and even-numbered *n*-alkanes with an unknown origin.

Comparison of the *n*-alcohol profiles to fresh beeswax (figure 7*a*) also shows distinct parallels. The *n*-alcohols in the ceramics are even-numbered and long-chain, consistent with the release of *n*-alcohols after hydrolysis of the palmitate wax esters (figure 7*b–e*). The source of the few short-chain and odd-numbered *n*-alcohols identified in the samples from Neve Yam, Tel Tsaf and Tsomet Shoket is unclear.

The long-chain even-numbered palmitate wax esters characteristic of fresh beeswax (figure 7*a*) [13] were also identified in the archaeological samples (figure 7*b–e*). Unusually, there seems to be a preferential loss of some of the longer wax esters, with the $C_{52}$ wax ester never preserved and the $C_{48}$ and $C_{50}$ wax esters absent or in low frequencies; this is seen most clearly in the samples from Neve Yam and Tsomet Shoket (figure 7*b,e*). The lack of the longer-chain wax esters in these samples may relate to their low abundance in fresh beeswax (figure 7*a*).

These patterns differ from plant waxes. Plants have similar *n*-alkane and *n*-alcohol profiles to beeswax [62,76], but the wax esters differ, comprising 32–64 carbon atoms [77], and some plant waxes may be further distinguished by the presence of non-palmitate and unsaturated wax esters (e.g. [78,79]). Such wax esters were not identified in the archaeological samples characterized as beeswax. Additional palmitate wax esters were identified in 15 other vessels, but in these vessels, the lipid yield was too low or the suite of markers present was insufficient to confidently identify beeswax. While these may represent highly degraded beeswax, some of these vessels contain exclusively wax esters with 24–38 carbon atoms that are suggestive of a different wax origin. These wax esters were identified in specific vessel forms including strainers, cornets and churns from Late Chalcolithic sites, with none identified in jars where beeswax markers were found.

It is highly significant that only four of the 247 vessels tested here (13.3% of the internal, crust and external samples with significant lipid yields) contained evidence for beeswax. This result is consistent with previous work carried out in the southern Levant, which identified few to no instances of lipids characteristic of beeswax preserved in pottery (table 3). Despite the large number of pottery vessels analysed from past and ongoing studies, totalling 467 vessels, beeswax residues were suggested for only 3% of the vessels tested or 35% of the vessels with over 5 µg g$^{-1}$ of lipids in the solvent extracted interiors, crusts and exteriors (figure 1*b* and table 3). This frequency is even lower if the identifications by Namdar [38] based exclusively on the *n*-alkane profiles are ignored (10% of samples with over 5 µg g$^{-1}$ of lipid). While the frequency of beeswax residue in sherds with significant lipid yields is higher than at some contemporary sites in Eurasia and North Africa [20], it certainly cannot be considered commonplace as it was identified at very few sites. The true frequency is also complicated by samples for which the lipid concentration was not calculated [81–83], and it may be masked by preservation biases.

# 4. Discussion and conclusion

The lipids preserved in the three jars and the crust from Neve Yam, Tel Tsaf, Tel Bene Beraq (South) and Tsomet Shoket mark some of the earliest evidence for beeswax use in the southern Levant. Based on the current study and a previous study [38], beeswax was already used throughout Israel in small amounts during the Late Chalcolithic period in relation to jars and potentially cornets. However, as we demonstrated above, beeswax utilization goes back to the Early and Middle Chalcolithic. While a confident biomolecular identification of degraded beeswax is possible, discerning the exact function or functions of beeswax is more difficult to establish. This is particularly true as the vessels under discussion appear to have been used alongside beeswax as well as animal fat and plant products. The specific use of beeswax is only easily identifiable when the vessel form is task-specific (e.g. clay hives and lamps). In this study, several suggestions can be put forward to explain the use of the beeswax in the tested vessels.

The identification of beeswax in only the exterior surface of the jar from Tsomet Shoket suggests that it was used as part of a post-firing treatment to alter the external vessel surface, possibly to form lustre. By rubbing it on a surface, polish is created, enhancing the vessel's aesthetics [1]. This technique is applied today to a variety of materials, including wood, leather, stone and textiles [1], and written sources show that this technique was used already during the Classical period in the Mediterranean; melted beeswax was mixed with various plant oils to form coatings on marble and wood to create shine [1, Pliny, XXXIII.122]. Such a function for ceramics is difficult to support because the visible wax would not survive diagenesis.

Beeswax could also be applied to ceramic vessels internally and externally as a sealant or to fill in cracks. Beeswax can be applied post-firing to make the vessel water resistant because of its

**Table 3.** Summary of ORA results obtained from 7th–4th cal. BC sites in the southern Levant and the number of vessels with suggested evidence for beeswax residues (sites tested for the current study are marked in bold).

| site | period | N vessels analysed | N vessels and crusts with > 5 µg g$^{-1}$ of lipids in the solvent extract internally | N vessels with > 5 µg g$^{-1}$ of lipids in the solvent extract exclusively externally | N vessels with evidence for beeswax | reference |
|---|---|---|---|---|---|---|
| Abu Thawaab | Pottery Neolithic | 10 | 0 | — | 0 | [80] |
| 'Dhra | Pottery Neolithic | 24 | 0 | — | 0 | [80] |
| Munhata | Pottery Neolithic | 13 | 0 | — | 0 | [80] |
| **Sha'ar HaGolan** | **Pottery Neolithic** | **16** | 0 | — | **0** | [80] |
| | | **16** | **1** | **1** | | **current study** |
| al-Basatin | Early Chalcolithic | 7 | 0 | — | 0 | [80] |
| Ein Zippori[a] | Early Chalcolithic | 3 | — | — | 0 | [81] |
| **Neve Yam** | **Early Chalcolithic** | **17** | **1** | — | **1** | **current study** |
| | | 33 | 0 | — | | [80] |
| **Tel Hreiz** | **Early Chalcolithic** | **3** | **1** | — | **0** | **current study** |
| Tabaqat al-Buma | Early Chalcolithic | 15 | 0 | — | 0 | [80] |
| **Tel Tsaf** | **Middle Chalcolithic** | **100** | **14** | **1** | **1** | **current study** |
| Tubna | Middle Chalcolithic | 22 | 0 | — | 0 | [80] |
| Abu Hof[a] | Late Chalcolithic | 10 | — | — | 0 | [82] |
| **Abu Matar** | **Late Chalcolithic** | **14** | 0 | — | **0** | [80] |
| | | **3** | **0** | **0** | | **current study** |
| **Ashqelon Agamim** | **Late Chalcolithic** | **28** | **1** | **0** | **0** | **current study** |
| **Azor** | **Late Chalcolithic** | **1** | **0** | **0** | **0** | **current study** |
| **Tel Bene Beraq (South)** | **Late Chalcolithic** | **10** | **1** | **1** | **1** | **current study** |
| **Bir es-Safadi** | **Late Chalcolithic** | **11** | **0** | **0** | **0** | **current study** |

(Continued.)

**Table 3.** (Continued.)

| site | period | N vessels analysed | N vessels and crusts with > 5 µg g⁻¹ of lipids in the solvent extract internally | N vessels with > 5 µg g⁻¹ of lipids in the solvent extract exclusively externally | N vessels with evidence for beeswax | reference |
|---|---|---|---|---|---|---|
| Ein Gedi | Late Chalcolithic | 10 | 6 | — | 6 | [38] |
| **Fazael 7** | **Late Chalcolithic** | **8** | **0** | **0** | **0** | **current study** |
| **Gilat** | **Late Chalcolithic** | **13** | **3** | **2** | **0** | **current study** |
| | | 10ᵃ | — | — | | [83] |
| Grar | Late Chalcolithic | 4 | 2 | — | 2 | [38] |
| Horvat Qarqar | Late Chalcolithic | 4 | 0 | — | 0 | [38] |
| Moringa Cave | Late Chalcolithic | 4 | 2 | — | 2 | [38] |
| Nahal Qomem | Late Chalcolithic | 6 | 0 | — | 0 | [38] |
| Nevatim | Late Chalcolithic | 4 | 0 | — | 0 | [80] |
| **Nisim Aloni** | **Late Chalcolithic** | **9** | **0** | **0** | **0** | **current study** |
| **Peqi'in** | **Late Chalcolithic** | **3** | **0** | **0** | **0** | **current study** |
| **Tsomet Shoket** | **Late Chalcolithic** | **18** | **0** | **1** | **1** | **current study** |
| Shiqmimᵃ | Late Chalcolithic | 11 | — | — | 0 | [82] |
| **Yehud** | **Late Chalcolithic** | **7** | **1** | **1** | **0** | **current study** |
| **Total** | | 467 | 33ᵇ | 7 | 14 | |

[a]The lipid concentration was not calculated, so preservation cannot be discussed.
[b]The number of samples with a TLE over 5 µg g⁻¹ cannot be calculated accurately as the TLE was not reported in some published studies [81–83].

hydrophobic nature [1]. This has been frequently suggested in ORA studies that identified beeswax [9,21,30,44,84]. A beeswax sealant would benefit vessels meant to store liquids, and correspondingly, beeswax biomarkers were identified in vessel forms related to storage and liquids—jars in this study and amphorae, juglets, flasks and chalices in previous studies [24,44,84]. Cooking vessels would likely not receive a beeswax sealant because beeswax melts at 62–65°C [15], so the sealant would need to be frequently reapplied, and the flavour and texture of the sealant could be undesirably incorporated into the cooked food.

Beeswax present in the interior surface of the two jars from Neve Yam and Tel Bene Beraq (South) and the crust from Tel Tsaf may additionally relate to the storage and use of unfiltered honey. The signature of filtered honey would be unlikely to preserve because honey is formed primarily from carbohydrates [85], which easily degrade [57]. Honey is a desirable sweetener [1] with a long history of use. Honey consumption in the wider area was documented as far back as 2300–1300 BC, with examples noted from Egyptian texts and the Bible (e.g. [1,39,86,87], Deuteronomy 32.13, Judges 14.8, Psalm 81.16. Samuel 14.25-27). Honey may also be used as a preservative for fruits, vegetables and even meat. This practice is documented back to the Classical period in the Roman cookbook – *Apicus* [88]. While these examples post-date the Chalcolithic period by several millennia, they suggest that honey had a clear role in the wider Mediterranean diet, and an earlier root is expected. Further, if the residue relates to the use of honey, then when the entire lipid profile is viewed together, we may suggest that honey was used in complex recipes with various plant and animal products. A purer and more intensive utilization can be suggested for the jar from Tel Bene Beraq (South) because the saturated fatty acid, *n*-alkane, *n*-alcohol and wax ester signatures more directly matches those of fresh beeswax (figure 7*a,d*).

Another possible function for beeswax identified in internal samples from Neve Yam, Tel Tsaf and Tel Bene Beraq (South) is the use of unfiltered honey in medicine. Honey is renowned for its anti-inflammatory and antibacterial properties and is frequently used in ointments [1,12,89]. Using honey in medicine in the Levant is documented in several papyri from Egypt (dated to *ca* 1900–1550 BC). They show that honey was used as an ointment, a laxative and as part of contraception [90,91]. Because medicinal use is based on textual sources, it cannot be linked to vessel forms. However, studies on vessels from significantly later archaeological sites in Europe identified ointments in miniature vessel forms [4,92,93]. The small vessel size likely relates to the ointment's value, and it is therefore unlikely that medicine would be stored in the vessels discussed in this paper, which include three comparatively large jars.

Vessel surface alteration and a dietary role specifically may be further supported by the co-occurrence of beeswax residue with other lipids such as ketones, which are often used to link archaeological samples to cooking (e.g. [94]) and post-firing surface treatments [60]. 16-hentriacontanone ($K_{31}$), 16-tritriacontanone ($K_{33}$) and 18-pentatriacontanone ($K_{35}$) form by heating palmitic and stearic acid to temperatures over 300°C or at lower temperatures for extended time [60,94]. The two potential stimuli for ketone synthesis are difficult to distinguish and are not mutually exclusive. However, in the prehistoric southern Levant, certain jars are often linked to cooking based on the occasional presence of soot marks on the exterior [52], and it is therefore possible that the jars from these sites were similarly used for cooking, melting beeswax or heating honey. It is also possible that the single ketone homologue ($K_{31}$) identified in the sample from Tel Tsaf relates to a plant wax; plant waxes are generally dominated by one homologue that is exclusively symmetrical [94].

Outside of diet and foodways, beeswax can be used as fuel for lighting [2]. This was suggested in previous ORA studies on zoomorphic vessels from Neolithic Greece [23] and lamps and conical cups from Late Minoan Crete [95]. These vessels are shallow with an open form, ideal for holding small contents of fuel. Analogous vessel forms were not identified in the prehistoric southern Levant [52], but a similar function was attributed to cornets based on ORA [38] (although the identification of beeswax is debatable). While the vessels are not shallow, the conical form is suitable for holding small amounts of wax. The vessels under discussion in this study have a much greater volume and depth and were likely unsuitable for lighting.

The use of beeswax during the Late Chalcolithic period was also mentioned in relation to copper production using the lost-wax technique [11]. Fragments of the moulds required for the technique have not yet been recovered from Chalcolithic sites. Therefore, while beeswax clearly played a role in copper production, at present without the moulds, the direct connection between beeswax and metallurgy cannot be discussed using ORA. Beeswax biomarkers were also so far not identified in pottery vessels analysed by GC-MS from sites with evidence for copper production (e.g. Abu Matar, Fazael and Shiqmim; table 3) [80,82,96,97], but this cannot be used to discredit beeswax's role in the industry at these sites.

Regardless of how bees and bee products were used, evidence from the lipid record suggests that bee products were used in low frequencies. Evidence for beeswax was only found in four out of the 247 vessels tested as part of this study (13.3% of vessels with significant lipid yields internally or externally) or, if the results of prior studies based on *n*-alkane profiles are accepted [38], as much as 14 out of the 467 vessels analysed by GC-MS (35% of vessels with significant lipid yields internally or externally) (table 3). A low utilization must be suggested cautiously because lipids poorly preserve in the southern Levant (e.g. table 3; [80,98,99]) and bee products may have been used and stored in organic containers [1,100]. Therefore, the lack of evidence does not necessarily imply lack of use. Nonetheless, the number of sites with beeswax residues in the southern Levant (7/30 tested sites) is too little to consider a potential widespread utilization in the region. The absence is particularly clear at Tel Tsaf, from which 100 ceramic vessels were tested, but only one sherd yielded evidence for beeswax (table 3). This absence is paralleled in the northern Levant at Tel Sabi Abyad, from which 287 vessels were sampled and none of the 41 vessels with significant lipid yields contained evidence for beeswax [101].

The limited use of bee products is noteworthy because they have numerous benefits, and even without beekeeping, wild bees could have been exploited (e.g. [102,103]). *Apis mellifera* was certainly present in the region as it has a wide distribution, found in wet, dry, cold and warm environments in Africa, Europe and parts of Asia [104]. Why then were beeswax and honey rarely used in the prehistoric southern Levant? This question is particularly important as only a few millennia later there is textual evidence for bee exploitation, and apiaries were identified at Tel Rehov [41], but the organic and archaeological records offer no clear root for this extensive bee exploitation.

A potential hypothesis to explain this low use is that beeswax exploitation contrasted with the Chalcolithic socio-economic system. The Chalcolithic period is characterized by agro-pastoralism [105], with minimal evidence for hunting, gathering or fishing, even when wild resources were readily accessible [106,107]. Following the same trend, people likely exploited wild bee products in low amounts. Despite the numerous functions and benefits of bee products, the Chalcolithic communities may have used alternatives that were more complementary to their socio-economic system. Alternatives can be suggested in ceramic vessel surface treatment and foodways.

As shown previously, beeswax can be applied to ceramics as a sealant or to form a shiny surface. During the Chalcolithic period, ceramics were altered primarily with paint applied with a brush or a slip or by burnishing [52], and there is at present no evidence for techniques that employed organic materials (e.g. [60]). Shine, when present, was created primarily by burnishing using another object [52]. Furthermore, because the Chalcolithic period is characterized by increasing standardization in craft production (e.g. [108,109]) and within this ceramic production (e.g. [49]), techniques that did not conform to the technological norm, such as the application of beeswax, were likely used by limited individuals. The example from Tsomet Shoket may represent an exceptional case where beeswax was used as part of a surface treatment, especially considering that the interior and exterior bear no visible surface treatments (figure 6).

Evidence for alternatives in the Chalcolithic cuisine are difficult to identify because of biases toward charred seeds in botanic preservation, but food could be seasoned with wild herbs and plants and sweetened with wild fruits (e.g. figs and dates; [110]). One must also consider that sweet foods, while now commonplace, were originally considered a luxury in various cultures [1,111]. Honey's value in the wider Levant is evident because many civilizations used it as an offering to gods. This practice can be dated as far back as 2450 BC in Sumer, 1567–1085 BC in Egypt and 700 BC in Israel [1, II Chronicles 31.5]. Considering its supposed value, honey was likely not part of the daily diet, although this must be suggested with caution because of preservation biases against carbohydrates—the main component of honey [57,85].

To conclude, the current study brings direct evidence for early use of beeswax and, by association, the likelihood of the use of honey during the Chalcolithic period of the southern Levant. These seemingly played a minor role, with the potential exception of the copper metallurgy industry. The exact function of the vessels where beeswax was found is difficult to identify because the various uses of bee products leave identical lipid signatures. However, based on the vessel forms and comparison to the wider Eastern Mediterranean record, it can be suggested that the vessels were used to contain beeswax or honey or that the beeswax was applied to their surface to alter its appearance and properties. Our results, bearing in mind preservation issues, allow us to cautiously suggest that bee products were not frequently produced and used (at least not in an organized fashion) during the Chalcolithic period. This was probably the situation until the Iron Age, during which apiaries were clearly in use. Thus, while the ceramic vessels tested here mark an early evidence for bee product

residues in ceramics, they by no means mark the root of extensive bee exploitation that appeared millennia after the fading of the Chalcolithic period. Further analysis of additional vessels from Bronze Age sites is required to elucidate the trajectory of bee exploitation in the region and identify when bees became an important part of the economy.

Data accessibility. All data and research materials supporting the results are in the article and electronic supplementary material.

Authors' contributions. R.C. conceptualized the study, carried out the laboratory work and data analysis and drafted the manuscript. D.R. helped conceptualize the study, draft the manuscript and acquisition funds. D.R., F.K., R.B., D.G., A.D. and E..G. provided the ceramic vessels for analysis. C.S. supervised the GC-MS analysis and helped conceptualize the study and draft the manuscript. All authors gave final approval for publication and agree to be held accountable for the work performed therein.

Competing interests. We declare we have no competing interests.

Funding. This work was supported by the ISF under grant no. 2016/2017; the Rust Family Foundation; the Irene Levi-Sala CARE Foundation; the Eurasia Department of the German Archaeological Institute (DAI) in Berlin; and the Zinman Institute of Archaeology, University of Haifa.

Acknowledgments. We would like to thank the Israel Antiquities Authority, E.C.M. van den Brink, T. Levy, Y. Garfinkel, S. Bar, Y. Abadi-Reiss, D. Varga and D. Shalem for allowing R. Chasan to analyse pottery found during their excavations for her PhD research. A special thanks to M. Belser and S. Cafisso for their assistance in the laboratory and S. Haad for her graphical assistance. We would also like to thank S. Buckley for his comments on an earlier version of this paper.

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
