## [Peer Review File · Royal Society Open Science]

Review History

RSOS-210950.R0 (Original submission)

Review form: Reviewer 1

Is the manuscript scientifically sound in its present form?

Yes

Are the interpretations and conclusions justified by the results?

Yes

Is the language acceptable?

Yes

Do you have any ethical concerns with this paper?

No

Have you any concerns about statistical analyses in this paper?

No

Recommendation?

Accept with minor revision (please list in comments)

Comments to the Author(s)

This paper reports on the investigation of beeswax in Chalcolithic pottery from the southern Levant using organic residue analysis methods. This extensive research, on almost 250 vessels, is very interesting as it presents one of the first unambiguous evidence for the exploitation of beehive products in this region. The interpretations are supported by the data and the different potential uses of beehive products are well discussed and documented. I therefore recommend that the article be accepted with minor corrections, after addressing the corrections in the attached file (see Appendix A).

Review form: Reviewer 2**Is the manuscript scientifically sound in its present form?**

Yes

Are the interpretations and conclusions justified by the results?

Yes

Is the language acceptable?

Yes

Do you have any ethical concerns with this paper?

No

Have you any concerns about statistical analyses in this paper?

No

Recommendation?

Accept with minor revision (please list in comments)

Comments to the Author(s)

This manuscript presents the evidence for the presence of beeswax in sherds from the Southern Levant using organic residue analyses. While the identification of beeswax in the samples seem rather robust (more on this below), the paper is discussing rather uncritically some previous identification of "beeswax" in the Levantine record. A discussion regarding previous identification should be included in the paper and the necessary changes made throughout to reflect the erroneous identification of beeswax.

I recommend this paper to be published with minor corrections (detailed below).

Ref [31] (Namdar et al. 2009, JAS) identifies putative beeswax in cornets from the Chalcolithic based on the distribution of n-alkanes. However, the sole presence of n-alkanes (when fatty acids / alcohols / wax esters are absent) cannot be seen as a proof for the presence of beeswax. Whelton et al. 2021 (JAS) have written about this recently in their "call for caution". This argument should be detailed in the manuscript as many of the published literature about "beeswax" in the Levant is not identifying beeswax in the way it should be. Those finds (such as ref 34 at line 62 or ref 35 at line 67; but there are others) should not be included in the list of sites with evidence for beeswax.

This comment about n-alkanes leads me to the structure of the paper. The introduction exposes the evidence for the exploitation of bee products in the region, including from organic residue

analyses. There is a need of a paragraph in the introduction discussing the suite of compounds that are characteristic of beeswax. This information comes right at the end – with the study of fresh beeswax – but this is too late, as the reader cannot relate the archaeological lipid assemblage to what is expected for beeswax and degraded beeswax.

Affiliations: to be double-checked as there are 8 listed affiliations but only 7 numbers by the authors' names.

Abstract: should be reworked based on my comments about the manuscript.

l. 44-45 "although the proportion is potentially biased by preservation conditions and sample size". This sentence is unclear – please rephrase.

l. 75. "Beeswax is rare" – this statement should be supported by numbers. How many sherds analysed? How many sherds with appreciable amount of lipids? How many beeswax residues?

Due to the poor preservation of lipids in the region, residues with significant amount of lipids are rare, but within that beeswax are rather common (?).

Based on table 1, 467 sherds have been analysed of which the authors have lipid data for 457.

Around 33 have > 5 ug/g of lipids (recovery rate of 7%). Beeswax has been evidenced in 14 vessels (this number will be lower when sherds with n-alkanes only will be removed). Beeswax is thus recorded in a whopping 42% of sherds!

l. 101. Worth mentioning in this paragraph the number of sherds that were analysed in this study; with number of sherds for each vessel type and each sites. That would allow the reader to grasp the full coverage of the study. Crusts should be mentioned here too.

l. 111. More details is needed about the internal and external surfaces, as this does not seem to be a standard protocol (no literature relating to the sampling method is cited). What depth? What area? How do they relate one to another? The authors seem to have analysed some crusts too – what is the sampling protocol for those?

l. 114. "Lipids from the external ceramic surface or soil samples were analysed as controls". Why? Add "to check for xxx".

l. 118. "to ensure that the 'bound' lipids were extracted without hydrolysis of the esterified lipids". Surely the main reason of using a solvent-extraction is to not hydrolyse hydrolyseable (WE and HWE) lipids? Could be rephrased "to ensure that solvent-soluble lipids were extracted without hydrolysis of the esterified lipids".

l. 120. "Lipids were extracted three times". Details are needed here regarding length of time for the sonication step.

l. 125. "Hexatriacontane (C₃₆ n-alkane; 98%, Sigma-Aldrich) was measured as an internal or external standard for TLE quantification". This does not make sense. Are the authors adding a known amount (how much?) of IS in their lipid extracts for quantification? Or are the extracts quantified using a known amount of compounds? Same comment for l. 133.

l.153. "Lipid preservation was challenging". This is not correct. Do the authors mean "low"?

l. 158. "which may relate to spillage or the application of a vessel treatment". It is worth elaborating on this idea a bit further. What would lead to absorbed residues being only deposited only in the outer surface? Cite some literature here too.

l. 161. "Among these" – please be more specific, as the previous paragraph talks about multiple extracts.

l. 167. As explained earlier – this should be amended as it refers to the identification of beeswax based solely on n-alkanes.

Table 1 is rather confusing (and should go into the SI as this is a repeat from the GC/MS chromatograms and the histograms from Fig 7). That would be good to visually group the 2 samples that are coming from the same sherd (outer/inner surface). "Monopalmitin" should read "monopalmitin". Should BB-12 and BB-21 be the same sample? What are Layers? What is the difference between "TSF18-57 In L1" and "TSF18-57 In L2"? This should be explained in the Methods section.

Comments about the four sherds with beeswax lipid assemblage:

NY-12 (from line 176) Cholesterol is known to oxidise rapidly in presence of fatty acids and clay (Hammann et al. 2018, Tetrahedron Letters). Given the very poor preservation of lipids at the site, the detection of cholesterol in that sample is rather suspicious – this could potentially indicate modern contamination.

Unsaturated fatty acids (C16:1, C18:1 and C20:1) are interpreted as arising from plant oils. The C18:1 is very likely to arise from carcass fats (very abundant in fresh carcass fats). The C16:1 and C20:1 are not labelled in the chromatogram (Fig 3) – are they abundant? It seems rather over-stretched for me to interpret those as arising from plant oils as they are also present in other foodstuffs (eg modern salmon). What is the interpretation for K31 and K33?

Even-chain alkanes are not detected in modern beeswax, so their presence here is rather puzzling (and would suggest contamination).

I would argue that there is only weak evidence for beeswax in this sample (lack for LCFAs, alkanes, weak evidence for WE etc) – the interpretation should be tentative.

TSF-18-57 (from line 202) This sample has a much stronger signature of degraded beeswax. Again, I am sceptical about the interpretation of plant oils re: C18:n fatty acids.

BB-21 (from line 226) Good evidence there too, although the alkane distribution is rather peculiar. The distribution of the WE/HWE is very characteristic and well-preserved.

TS-4 (from line 249) Rather high even-numbered alkanes in this sample. The WE distribution is not very characteristic and well-preserved. The presence of beeswax is only tentative.

The chromatograms (Fig. 3-6) are rather difficult to read, as the amount of labels makes the chromatogram very busy and because of the thickness of the line indicating peaks of small size (eg for cholesterol, C15:0 and A16 in Fig 3) – it looks like the peak is much bigger than it is. I would suggest to use symbols and colours to indicate compound classes – with symbols just above the peak – that allows to see at first glance the distribution of compounds within compound classes (eg see Roffet-Salque et al. 2015, Nature).

l. 259. The lipid distribution of beeswax is species-specific (see Aichholz et al. 1999, J of Chromatography A) – so it is particularly important to provide details regarding the species (I assume *A. mellifera*).

l. 286 High molecular weight WE are usually well-preserved, with low-molecular weight WE degrading first. Could their absence in the extract be due to the GC-MS setup?

l. 298 Four out of 247 vessels contained beeswax. This is interesting but lipid preservation is rather low. How many sherds with appreciable amount of lipids? You should compare that number to the number of beeswax.

l. 306 “calculated TLE”. What does this mean? The lipid concentration was not calculated? (same in table 1 caption).

l. 317 Cornets here again.

l. 338 Evidence for waterproofing collared flasks with beeswax has also been shown by Salque et al 2013 (Nature).

l. 354 The authors use the presence of beeswax as potential evidence for propolis. What is the chemical composition of propolis? Does it fit with this interpretation?

l. 379 Misidentification of beeswax in cornets here again.

l. 391 See my comment for l. 75. Results should be reported against number of sherds with archaeological residues, rather than total number of sherds.

l. 400 For Tell Sabi Abyad, 287 vessels were sampled (the citation to ref 75 is erroneous and should be ref 95) but only 41 led to significant amount of archaeological lipids. So no beeswax in 41 sherds with lipids.

l. 403 Not sure “easily” is the right term for exploiting wild bees, given the difficulties honey hunters usually face to gather beeswax and honey.

l. 405 I see that the discussion is building on the low frequency of beeswax in the Levant, however, as I said earlier, this is hampered by the lack of residues. The recovery rate based on sherds with archaeological lipids is much higher, and probably not far off from other regions... I would advise the discussion section to be re-phrased to take this into account.

Figure 1. Panel A is an update from a figure from ref 17. Panel B is very relevant to the paper. However, it is rather misleading as many of the sites did not lead sherds with significant amount of lipids. This figure does thus not allow to really see what is happening in the region (given that the authors are arguing for low intensity). This information could be added – nb of sherds analysed, nb of sherds with lipids, nb of sherds with beeswax (or percentages) maybe using pie charts? That way Table 2 could move to SI.

References. Please check the references from misspelling etc as there seem to be some issues. Eg Ref 70 should read “Eglinton”

Decision letter (RSOS-210950.R0)

Dear Dr Chasan

The Editors assigned to your paper RSOS-210950 "Bee products in the prehistoric southern Levant: evidence from the lipid organic record" have now received comments from reviewers and would like you to revise the paper in accordance with the reviewer comments and any comments from the Editors. Please note this decision does not guarantee eventual acceptance.

You will see that both referees express positive views on the paper, but each recommends a number of changes before publication. Whilst the referees individually summarise their opinion as that the paper requires minor revision, taking their comments together the recommended revisions are quite substantial and the editorial decision is therefore that the paper requires major revision -- this will allow you a little more time to address the referees' comments thoroughly.

Please submit your revised manuscript and required files (see below) no later than 21 days from today's (ie 09-Aug-2021) date. Note: the ScholarOne system will 'lock' if submission of the revision is attempted 21 or more days after the deadline. If you do not think you will be able to meet this deadline please contact the editorial office immediately.

on behalf of Dr Michelle Alexander (Associate Editor) and Peter Haynes (Subject Editor)
openscience@royalsociety.org

Associate Editor Comments to Author (Dr Michelle Alexander):

Comments to the Author:

The reviewers agree that the paper is worthy of scientific merit. Taken together, the revisions requested are rather substantial and in keeping with major revisions. Particularly note their suggestions to reframe the discussion, review previous identification of beeswax more critically and provide greater clarity in the methods and presentation of the results.

Reviewer comments to Author:

Reviewer: 1

Comments to the Author(s)

This paper reports on the investigation of beeswax in Chalcolithic pottery from the southern Levant using organic residue analysis methods. This extensive research, on almost 250 vessels, is very interesting as it presents one of the first unambiguous evidence for the exploitation of beehive products in this region. The interpretations are supported by the data and the different potential uses of beehive products are well discussed and documented. I therefore recommend that the article be accepted with minor corrections, after addressing the corrections in the attached file

Reviewer: 2

Comments to the Author(s)

This manuscript presents the evidence for the presence of beeswax in sherds from the Southern Levant using organic residue analyses. While the identification of beeswax in the samples seem rather robust (more on this below), the paper is discussing rather uncritically some previous identification of "beeswax" in the Levantine record. A discussion regarding previous identification should be included in the paper and the necessary changes made throughout to reflect the erroneous identification of beeswax.

I recommend this paper to be published with minor corrections (detailed below).

Ref [31] (Namdar et al. 2009, JAS) identifies putative beeswax in cornets from the Chalcolithic based on the distribution of n-alkanes. However, the sole presence of n-alkanes (when fatty acids / alcohols / wax esters are absent) cannot be seen as a proof for the presence of beeswax. Whelton et al. 2021 (JAS) have written about this recently in their "call for caution". This argument should be detailed in the manuscript as many of the published literature about "beeswax" in the Levant is not identifying beeswax in the way it should be. Those finds (such as ref 34 at line 62 or ref 35 at line 67; but there are others) should not be included in the list of sites with evidence for beeswax.

This comment about n-alkanes leads me to the structure of the paper. The introduction exposes the evidence for the exploitation of bee products in the region, including from organic residue

analyses. There is a need of a paragraph in the introduction discussing the suite of compounds that are characteristic of beeswax. This information comes right at the end – with the study of fresh beeswax – but this is too late, as the reader cannot relate the archaeological lipid assemblage to what is expected for beeswax and degraded beeswax.

Affiliations: to be double-checked as there are 8 listed affiliations but only 7 numbers by the authors' names.

Abstract: should be reworked based on my comments about the manuscript.

l. 44-45 "although the proportion is potentially biased by preservation conditions and sample size". This sentence is unclear – please rephrase.

l. 75. "Beeswax is rare" – this statement should be supported by numbers. How many sherds analysed? How many sherds with appreciable amount of lipids? How many beeswax residues?

Due to the poor preservation of lipids in the region, residues with significant amount of lipids are rare, but within that beeswax are rather common (?).

Based on table 1, 467 sherds have been analysed of which the authors have lipid data for 457.

Around 33 have > 5 ug/g of lipids (recovery rate of 7%). Beeswax has been evidenced in 14 vessels (this number will be lower when sherds with n-alkanes only will be removed). Beeswax is thus recorded in a whopping 42% of sherds!

l. 101. Worth mentioning in this paragraph the number of sherds that were analysed in this study; with number of sherds for each vessel type and each sites. That would allow the reader to grasp the full coverage of the study. Crusts should be mentioned here too.

l. 111. More details is needed about the internal and external surfaces, as this does not seem to be a standard protocol (no literature relating to the sampling method is cited). What depth? What area? How do they relate one to another? The authors seem to have analysed some crusts too – what is the sampling protocol for those?

l. 114. "Lipids from the external ceramic surface or soil samples were analysed as controls". Why? Add "to check for xxx".

l. 118. "to ensure that the 'bound' lipids were extracted without hydrolysis of the esterified lipids". Surely the main reason of using a solvent-extraction is to not hydrolyse hydrolyseable (WE and HWE) lipids? Could be rephrased "to ensure that solvent-soluble lipids were extracted without hydrolysis of the esterified lipids".

l. 120. "Lipids were extracted three times". Details are needed here regarding length of time for the sonication step.

l. 125. "Hexatriacontane (C₃₆ n-alkane; 98%, Sigma-Aldrich) was measured as an internal or external standard for TLE quantification". This does not make sense. Are the authors adding a known amount (how much?) of IS in their lipid extracts for quantification? Or are the extracts quantified using a known amount of compounds? Same comment for l. 133.

l.153. "Lipid preservation was challenging". This is not correct. Do the authors mean "low"?

l. 158. "which may relate to spillage or the application of a vessel treatment". It is worth elaborating on this idea a bit further. What would lead to absorbed residues being only deposited only in the outer surface? Cite some literature here too.

l. 161. "Among these" – please be more specific, as the previous paragraph talks about multiple extracts.

l. 167. As explained earlier – this should be amended as it refers to the identification of beeswax based solely on n-alkanes.

Table 1 is rather confusing (and should go into the SI as this is a repeat from the GC/MS chromatograms and the histograms from Fig 7). That would be good to visually group the 2 samples that are coming from the same sherd (outer/inner surface). "Monopalmitin" should read "monopalmitin". Should BB-12 and BB-21 be the same sample? What are Layers? What is the difference between "TSF18-57 In L1" and "TSF18-57 In L2"? This should be explained in the Methods section.

Comments about the four sherds with beeswax lipid assemblage:

NY-12 (from line 176) Cholesterol is known to oxidise rapidly in presence of fatty acids and clay (Hammann et al. 2018, Tetrahedron Letters). Given the very poor preservation of lipids at the site, the detection of cholesterol in that sample is rather suspicious – this could potentially indicate modern contamination.

Unsaturated fatty acids (C16:1, C18:1 and C20:1) are interpreted as arising from plant oils. The C18:1 is very likely to arise from carcass fats (very abundant in fresh carcass fats). The C16:1 and C20:1 are not labelled in the chromatogram (Fig 3) – are they abundant? It seems rather over-stretched for me to interpret those as arising from plant oils as they are also present in other foodstuffs (eg modern salmon). What is the interpretation for K31 and K33?

Even-chain alkanes are not detected in modern beeswax, so their presence here is rather puzzling (and would suggest contamination).

I would argue that there is only weak evidence for beeswax in this sample (lack for LCFAs, alkanes, weak evidence for WE etc) – the interpretation should be tentative.

TSF-18-57 (from line 202) This sample has a much stronger signature of degraded beeswax. Again, I am sceptical about the interpretation of plant oils re: C18:n fatty acids.

BB-21 (from line 226) Good evidence there too, although the alkane distribution is rather peculiar. The distribution of the WE/HWE is very characteristic and well-preserved.

TS-4 (from line 249) Rather high even-numbered alkanes in this sample. The WE distribution is not very characteristic and well-preserved. The presence of beeswax is only tentative.

The chromatograms (Fig. 3-6) are rather difficult to read, as the amount of labels makes the chromatogram very busy and because of the thickness of the line indicating peaks of small size (eg for cholesterol, C15:0 and A16 in Fig 3) – it looks like the peak is much bigger than it is. I would suggest to use symbols and colours to indicate compound classes – with symbols just above the peak – that allows to see at first glance the distribution of compounds within compound classes (eg see Roffet-Salque et al. 2015, Nature).

l. 259. The lipid distribution of beeswax is species-specific (see Aichholz et al. 1999, J of Chromatography A) – so it is particularly important to provide details regarding the species (I assume *A. mellifera*).

l. 286 High molecular weight WE are usually well-preserved, with low-molecular weight WE degrading first. Could their absence in the extract be due to the GC-MS setup?

l. 298 Four out of 247 vessels contained beeswax. This is interesting but lipid preservation is rather low. How many sherds with appreciable amount of lipids? You should compare that number to the number of beeswax.

l. 306 “calculated TLE”. What does this mean? The lipid concentration was not calculated? (same in table 1 caption).

l. 317 Cornets here again.

l. 338 Evidence for waterproofing collared flasks with beeswax has also been shown by Salque et al 2013 (Nature).

l. 354 The authors use the presence of beeswax as potential evidence for propolis. What is the chemical composition of propolis? Does it fit with this interpretation?

l. 379 Misidentification of beeswax in cornets here again.

l. 391 See my comment for l. 75. Results should be reported against number of sherds with archaeological residues, rather than total number of sherds.

l. 400 For Tell Sabi Abyad, 287 vessels were sampled (the citation to ref 75 is erroneous and should be ref 95) but only 41 led to significant amount of archaeological lipids. So no beeswax in 41 sherds with lipids.

l. 403 Not sure “easily” is the right term for exploiting wild bees, given the difficulties honey hunters usually face to gather beeswax and honey.

l. 405 I see that the discussion is building on the low frequency of beeswax in the Levant, however, as I said earlier, this is hampered by the lack of residues. The recovery rate based on sherds with archaeological lipids is much higher, and probably not far off from other regions... I would advise the discussion section to be re-phrased to take this into account.

Figure 1. Panel A is an update from a figure from ref 17. Panel B is very relevant to the paper. However, it is rather misleading as many of the sites did not lead sherds with significant amount of lipids. This figure does thus not allow to really see what is happening in the region (given that the authors are arguing for low intensity). This information could be added – nb of sherds analysed, nb of sherds with lipids, nb of sherds with beeswax (or percentages) maybe using pie charts? That way Table 2 could move to SI.

References. Please check the references from misspelling etc as there seem to be some issues. Eg Ref 70 should read “Eglinton”

===PREPARING YOUR MANUSCRIPT===

===PREPARING YOUR REVISION IN SCHOLARONE===

Please ensure that you include a summary of your paper at Step 2 'Type, Title, & Abstract'. This should be no more than 100 words to explain to a non-scientific audience the key findings of your

research. This will be included in a weekly highlights email circulated by the Royal Society press office to national UK, international, and scientific news outlets to promote your work.

Author's Response to Decision Letter for (RSOS-210950.R0)

See Appendix B.

Decision letter (RSOS-210950.R1)

Dear Dr Chasan,

It is a pleasure to accept your manuscript entitled "Bee products in the prehistoric southern Levant: evidence from the lipid organic record" in its current form for publication in Royal Society Open Science. There were no reviewers at this stage but the Associate Editor is satisfied that the authors have diligently addressed all of the previous comments and requirements from the two reviewers.

on behalf of Dr Michelle Alexander (Associate Editor) and Peter Haynes (Subject Editor)
openscience@royalsociety.org

Appendix A

This paper reports on the investigation of beeswax in Chalcolithic pottery from the southern Levant using organic residue analysis methods. This extensive research, on almost 250 vessels, is very interesting as it presents one of the first unambiguous evidence for the exploitation of beehive products in this region. The interpretations are supported by the data and the different potential uses of beehive products are well discussed and documented. I therefore recommend that the article be accepted with minor corrections, after addressing the following:

Fig. 1: some published data are not represented. Please add data from Casanova et al., 2020; Drieu et al., 2021; Heron et al., 2015; Rageot et al., 2021; Stojanovski et al., 2020.

Throughout the article, the simple citation of the following articles: Mazar et al., 2008, Namdar et al., 2009, Namdar et al., 2017 and Amir et al., 2021 is a problem, as the biomarkers that are used to support the identification of beeswax are not those of beeswax (see for example Whelton et al., 2021). The experiments set up to support the identification of beeswax are not detailed enough and the primary data is not available, so the conclusions reached by the authors remain to be confirmed. As it is difficult to omit citing these papers given the topic of this manuscript, I suggest that wherever these papers are mentioned, the text should be modified, for example to read 'beeswax may have been identified' or 'the authors report/claim finding beeswax'.

End of the introduction (l. 94-96): It is strange to give the results in the introduction, without having presented the methods and data first. Please delete this part of the introduction.

Materials and methods: please be more specific about the list of samples analysed (this can be in the form of a table): list of sites, chronology of sites, number of samples per site, type of vessels studied at each site, type of sample (potsherd, crust)

l. 114-115: please elaborate on the soil samples: is it the soil that was found in the ceramics? on which the ceramics were found? the soil of the square / layer where the ceramics were found?

l. 115-117: There are several extraction methods presented in Stern et al. 2000. To guide the reader, you should specify which method you are referring to.

l. 117-119: the purpose of the sodium hydroxide solution is precisely to hydrolyse the saponifiable lipids. By definition, the extraction of the 'bound' lipids therefore results in hydrolysis. Please correct.

l. 125-126: Please specify when the internal standard was added. Can you justify why you have chosen to use an external standard? It is important to mention here for non-chemist readers that the use of an external standard does not allow for the evaluation of the extraction efficiency.

l. 156: the presence and analysis of a crust was never mentioned in the Materials and Methods. Please add this information.

l. 157-158: It is specified on l. 114-115 that the external surfaces were analysed as a control. If this is how they are interpreted, extraction yields $> 5 \mu\text{g/g}$ should be considered as contamination, and the samples discarded from the interpretation. It is indeed possible that the external surfaces have accumulated more lipids than the internal surfaces, either through overflow of the contents, application of a surface treatment or simply significant penetration of the products into the walls. If this is how the data is interpreted, the collection and analysis of the external walls should not be reported as control samples in the Materials and Methods section.

Results: In general, references to plant products should be removed. As the authors mention, mono- and polyunsaturated fatty acids also exist in animal fats, and given the amounts preserved in the

samples presented, there is no way of telling whether these originate from degraded plant oils or animal fats.

I. 161-164: For an uninformed reader, it is necessary to list the beeswax biomarkers that were detected in the 4 vessels. It is not sufficient to refer to Table 1, as it also includes other compounds that are not derived from beeswax. An alternative option could be to start the results section with the description of the biomarkers of fresh beeswax (I. 258-297).

Table 1: The different types of samples E, In L1, In L2, are not easy to understand and relate to the description in the Materials and Methods. In this section, for example, there is no mention of sampling two inner layers. Please elaborate on the description of the sampling in the Materials and Methods section. A figure with a drawing would be welcome, for example. In addition, adding a 'Vessel' column at the beginning of the table would make it easier to see which samples come from the same vessel, and thus to compare them. Bolding the beeswax biomarkers could also guide the reader unfamiliar with organic residue analysis.

I. 195-196: Please comment on the presence of the ketones K31 and K33 in the sample. In particular, the presence of markers of the heating of fatty products in this vessel should be discussed in relation to the preservation of the beeswax signal (heating of beeswax? e.g. Regert et al., 2001)

I. 215: again, please elaborate on the interpretation of the ketone.

I. 239-242: The presence of long chain dicarboxylic acids is very rarely mentioned in the literature. It would be useful to have a chromatogram showing these acids so that the reader can get an idea of the retention times and peak intensities in relation to the rest of the signal. Please explain in more detail why they are probably not oxidation products.

I. 259-260: Please clarify whether this section deals with fresh beeswax data published in the literature or whether you have analysed fresh beeswax. In the latter case, add this item in the Materials and Methods section

I. 265-266: 'In two of the vessels tested here and the acid fractions'. Please rephrase

I. 266-270: Does this mean that the bound fatty acids are protected from sublimation?

I. 286-287: 'Unusually, there seems to be a preferential loss of some of the longer wax esters, although this may also relate to their low abundance in fresh beeswax'. What is it about? Some ceramic samples? If so, which ones? Please rephrase.

I. 294-295: Short chain esters are used in some cosmetics, for example, myristic acid esters (Becker et al., 2010)

I.305-307: It should also be added that in many publications, the full range of beeswax biomarkers required for reliable identification (Whelton et al., 2021) is not present. The number of reliable identifications is therefore even smaller than reported.

I. 353-355: Propolis has a specific composition that includes terpenes, which do not seem to have been detected here. Elaborate a bit on the probability of dealing with propolis here, or delete the mention.

Discussion part, in particular I. 363-374: there is very little mention of the concomitant presence of fats, especially animal fats in most of the vessels where beeswax was detected. This is a fundamental aspect to consider when discussing the function of the containers. Please expand.

Still in discussion: from Figure 7, a clear distinction can be seen between BB-21 L2, whose beeswax profile is particularly well preserved, and the other 3 samples, whose profile is much more altered. Without ruling out the possibility of differential degradation, is it possible to discuss a potential different use of beeswax in these two groups of ceramics?

I. 391-401: it cannot be excluded that the beehive products were exploited in perishable containers rather than in ceramics. Please include this hypothesis to the discussion.

Chromatograms of the extractions of the bound fraction and of all control samples are not presented in the article. To meet the publication requirements of the journal, they should be a second piece of Supplementary Information.

References:

- Becker, Lillian C., Wilma F. Bergfeld, Donald V. Belsito, Ronald A. Hill, Curtis D. Klaassen, James G. Marks, Ronald C. Shank, Thomas J. Slaga, Paul W. Snyder, F. Alan Andersen, 2010. Final report of the amended safety assessment of myristic acid and its salts and esters as used in cosmetics. *International journal of toxicology* 29, 162S-186S
- Casanova, E., Arbogast, R.-M., Denaire, A., Jeunesse, C., Lefranc, P., Evershed, R.P., 2020. Spatial and temporal disparities in human subsistence in the Neolithic Rhineland gateway. *J. Archaeol. Sci.* 122, 105215. <https://doi.org/10.1016/j.jas.2020.105215>
- Drieu, L., Lucquin, A., Cassard, L., Sorin, S., Craig, O.E., Binder, D., Regert, M., 2021. A Neolithic without dairy? Chemical evidence from the content of ceramics from the Pendimoun rock-shelter (Castellar, France, 5750-5150 BCE). *J. Archaeol. Sci. Rep.* 35, 102682. <https://doi.org/10.1016/j.jasrep.2020.102682>
- Heron, C., Craig, O.E., Luquin, A., Steele, V.J., Thompson, A., Piličiauskas, G., 2015. Cooking fish and drinking milk? Patterns in pottery use in the southeastern Baltic, 3300-2400 Cal BC. *J. Archaeol. Sci.* 63, 33–43. <https://doi.org/10.1016/j.jas.2015.08.002>
- Rageot, M., Lepère, C., Henry, A., Binder, D., Davtian, G., Filippi, J.-J., Fernandez, X., Guilaine, J., Jallet, F., Radi, G., Thirault, E., Terradas, X., Regert, M., 2021. Management systems of adhesive materials throughout the Neolithic in the North-West Mediterranean. *J. Archaeol. Sci.* 126, 105309. <https://doi.org/10.1016/j.jas.2020.105309>
- Regert, M., Colinart, S., Degrand, L., Decavallas, O., 2001. Chemical alteration and use of beeswax through time: accelerated ageing tests and analysis of archaeological samples from various environmental contexts. *Archaeometry* 43, 549–569. <https://doi.org/10.1111/1475-4754.00036>
- Stojanovski, D., Živaljević, I., Dimitrijević, V., Dunne, J., Evershed, R.P., Balasse, M., Dowle, A., Hendy, J., McGrath, K., Fischer, R., Speller, C., Jovanović, J., Casanova, E., Knowles, T., Balj, L., Naumov, G., Putica, A., Starović, A., Stefanović, S., 2020. Living off the land: Terrestrial-based diet and dairying in the farming communities of the Neolithic Balkans. *PLOS ONE* 15, e0237608. <https://doi.org/10.1371/journal.pone.0237608>
- Whelton, H.L., Hammann, S., Cramp, L.J.E., Dunne, J., Roffet-Salque, M., Evershed, R.P., 2021. A call for caution in the analysis of lipids and other small biomolecules from archaeological contexts. *J. Archaeol. Sci.* 132, 105397. <https://doi.org/10.1016/j.jas.2021.105397>

Appendix B

Dear Royal Society Open Science editors,

We thank you and the reviewers for the comments on our paper “**Bee products in the prehistoric southern Levant: evidence from the lipid organic record.**” We carefully went through all the comments and suggestions made by the reviewers, and we revised the text in accordance to the reviewers’ suggestions. Our detailed point to point account of the changes made and answers to the reviewer’s comments are listed below. We hope this will suffice, and we will be happy to clarify any points that need further clarification.

Best wishes

Rivka Chasan

Laboratory for Ground Stone Tools Research

Zinman Institute of Archaeology, University of Haifa,

199 Abba Khousy Ave. Mount Carmel, Haifa, 3498838, Israel

Reviewer 1:

This paper reports on the investigation of beeswax in Chalcolithic pottery from the southern Levant using organic residue analysis methods. This extensive research, on almost 250 vessels, is very interesting as it presents one of the first unambiguous evidence for the exploitation of beehive products in this region. The interpretations are supported by the data and the different potential uses of beehive products are well discussed and documented. I therefore recommend that the article be accepted with minor corrections, after addressing the following:

Fig. 1: some published data are not represented. Please add data from Casanova et al., 2020; Drieu et al., 2021; Heron et al., 2015; Rageot et al., 2021; Stojanovski et al., 2020.

We thank the reviewer for pointing out these missing references. The sites have been added to figure 1 and the reference list.

Throughout the article, the simple citation of the following articles: Mazar et al., 2008, Namdar et al., 2009, Namdar et al., 2017 and Amir et al., 2021 is a problem, as the biomarkers that are used to support the identification of beeswax are not those of beeswax (see for example Whelton et al., 2021). The experiments set up to support the identification of beeswax are not detailed enough and the primary data is not available, so the conclusions reached by the authors remain to be confirmed. As it is difficult to omit citing these papers given the topic of this manuscript, I suggest that wherever these papers are mentioned, the text should be modified, for example to read 'beeswax may have been identified' or 'the authors report/claim finding beeswax'.

Based on the limited subset of lipid biomarkers in the vessels presented by those papers, we agree with the reviewer completely and have incorporated the suggested change so that the interpretations are presented more cautiously.

End of the introduction (l. 94-96): It is strange to give the results in the introduction, without having presented the methods and data first. Please delete this part of the introduction.

We thank the reviewer for this comment. As suggested, the ending of the introduction has been modified.

Materials and methods: please be more specific about the list of samples analysed (this can be in the form of a table): list of sites, chronology of sites, number of samples per site, type of vessels studied at each site, type of sample (potsherd, crust)

We thank the reviewer for this comment. As suggested, we added in an additional table detailing the vessels sampled and analysed by the leading author in the course of the wider study.

l. 114-115: please elaborate on the soil samples: is it the soil that was found in the ceramics? On which the ceramics were found? the soil of the square / layer where the ceramics were found?

We thank the reviewer for this comment. Soil was sampled from the layers the ceramics were retrieved from in close spatial proximity. This has been clarified in the text.

l. 115-117: There are several extraction methods presented in Stern et al. 2000. To guide the reader, you should specify which method you are referring to.

We thank the reviewer for this comment. Accordingly, we have specified that the extraction followed a modified version of Method D (saponification) outlined by Stern et al. 2000.

l. 117-119: the purpose of the sodium hydroxide solution is precisely to hydrolyse the saponifiable lipids. By definition, the extraction of the 'bound' lipids therefore results in hydrolysis. Please correct.

We thank the reviewer for this comment. The reason for this two-step extraction technique has been clarified. Solvent extraction first allowed all wax esters and acylglycerols to be studied, and the following saponification allowed any bound lipids to be analysed.

l. 125-126: Please specify when the internal standard was added. Can you justify why you have chosen to use an external standard? It is important to mention here for non-chemist readers that the use of an external standard does not allow for the evaluation of the extraction efficiency.

An internal standard (tetratriacontane) was not used because early analysis showed that the C₃₄ *n*-alkane co-eluted with some key molecules, deterring accurate quantification. A similar problem occurred with hexatriacontane when it was added to the samples after silylation or methylation; because of this, hexatriacontane was run in the GC-MS sequence after every

four samples to allow for a more accurate quantification. This later point has been clarified in the text.

l. 156: the presence and analysis of a crust was never mentioned in the Materials and Methods. Please add this information.

This information has been added in the methodology section.

l. 157-158: It is specified on l. 114-115 that the external surfaces were analysed as a control. If this is how they are interpreted, extraction yields $> 5 \mu\text{g/g}$ should be considered as contamination, and the samples discarded from the interpretation. It is indeed possible that the external surfaces have accumulated more lipids than the internal surfaces, either through overflow of the contents, application of a surface treatment or simply significant penetration of the products into the walls. If this is how the data is interpreted, the collection and analysis of the external walls should not be reported as control samples in the Materials and Methods section.

We thank the reviewer for this comment and agree that externally identified significant lipid yields are problematic to interpret. This is particularly true for the related pottery, which is low-fired and has a high-porosity, allowing lipids to more easily migrate. Because of this, differentiating post-depositional contamination using external ceramic controls is difficult. We chose to include the samples with significant lipid yields only externally because if these were related to contamination, then a higher lipid yield should also be identified internally. This point has been clarified in the text.

Results: In general, references to plant products should be removed. As the authors mention, mono and polyunsaturated fatty acids also exist in animal fats, and given the amounts preserved in the samples presented, there is no way of telling whether these originate from degraded plant oils or animal fats.

We thank the reviewer for this comment. Instead of removing references to plant products, the suggestion has been presented with greater caution.

l. 161-164: For an uninformed reader, it is necessary to list the beeswax biomarkers that were detected in the 4 vessels. It is not sufficient to refer to Table 1, as it also includes other compounds

that are not derived from beeswax. An alternative option could be to start the results section with the description of the biomarkers of fresh beeswax (l. 258-297).

We thank the reviewer for this comment. The beeswax biomarkers are now presented in the beginning of the introduction.

Table 1: The different types of samples E, In L1, In L2, are not easy to understand and relate to the description in the Materials and Methods. In this section, for example, there is no mention of sampling two inner layers. Please elaborate on the description of the sampling in the Materials and Methods section. A figure with a drawing would be welcome, for example. In addition, adding a 'Vessel' column at the beginning of the table would make it easier to see which samples come from the same vessel, and thus to compare them. Bolding the beeswax biomarkers could also guide the reader unfamiliar with organic residue analysis.

We thank the reviewer for this comment. Accordingly, table 1 (which is now table 2) has been clarified, and it is specified that the initial ceramic surface layer was not analysed because of its high potential for contamination.

l. 195-196: Please comment on the presence of the ketones K31 and K33 in the sample. In particular, the presence of markers of the heating of fatty products in this vessel should be discussed in relation to the preservation of the beeswax signal (heating of beeswax? e.g. Regert et al., 2001)

l. 215: again, please elaborate on the interpretation of the ketone.

We thank the reviewer for this comment and have chosen to elaborate on the presence of ketones as they relate to vessel function and beeswax in the discussion. We believe that including the interpretation in the results would make the text read as repetitive.

l. 239-242: The presence of long chain dicarboxylic acids is very rarely mentioned in the literature. It would be useful to have a chromatogram showing these acids so that the reader can get an idea of the retention times and peak intensities in relation to the rest of the signal. Please explain in more detail why they are probably not oxidation products.

The chromatogram highlighting these long-chain dicarboxylic acids was added to the supplementary material (Supplementary material 2) so that it can be used by others to aid

in future identification. Long-chain dicarboxylic acids are rarely oxidation products because very long-chain unsaturated fatty acids are required to produce these and such molecules are uncommon.

1. 259-260: Please clarify whether this section deals with fresh beeswax data published in the literature or whether you have analysed fresh beeswax. In the latter case, add this item in the Materials and Methods section

We thank the reviewer for this comment. The samples were compared to published lipid profiles of fresh beeswax, and this point has been further clarified.

1. 265-266: 'In two of the vessels tested here and the acid fractions'. Please rephrase

We thank the reviewer for this comment. We have clarified that C_{16:0} is highly abundant in two of the solvent extracts and all of the acid fractions.

1. 266-270: Does this mean that the bound fatty acids are protected from sublimation?

We thank the reviewer for this comment. In general, as mentioned in the beginning of the results section, because of the alkaline soils and Mediterranean climate characteristic of Israel, lipids in the unbound fraction are nearly always poorly preserved. Therefore, preservation biases cannot be linked entirely to molecule chain-length. The bound lipids do however appear to be more protected. The connection between molecule preservation and sublimation has therefore been suggested more cautiously.

1. 286-287: 'Unusually, there seems to be a preferential loss of some of the longer wax esters, although this may also relate to their low abundance in fresh beeswax'. What is it about? Some ceramic samples? If so, which ones? Please rephrase.

We thank the reviewer for this comment. The sentence has been further clarified to explain that the C₅₂ wax ester was never identified and that the C₄₈ and C₅₀ wax esters were absent or in low frequencies in the samples. The vessels from Neve Yam and Tsomet Shoket were listed as examples.

1. 294-295: Short chain esters are used in some cosmetics, for example, myristic acid esters (Becker et al., 2010).

We thank the reviewer for this comment; however, the short- and medium-chain wax esters mentioned in the text are all palmitate and therefore cannot be related to the lipids identified in cosmetics.

1.305-307: It should also be added that in many publications, the full range of beeswax biomarkers required for reliable identification (Whelton et al., 2021) is not present. The number of reliable identifications is therefore even smaller than reported.

We thank the reviewer for this comment. Over the course of this study, we were very cautious with what residues we defined as beeswax, and we always considered alternative origins for these molecules. The four vessels in this paper present the most convincing lipid profiles, although they are not always perfect matches. This can be related to preservation biases. The less clear examples are now presented with greater caution. Earlier studies on Israeli pottery however do seem prone to over interpretation, and they characterised beeswax based only on the *n*-alkane profiles. These earlier identifications are now presented in the text in the results and introduction with more caution.

l. 353-355: Propolis has a specific composition that includes terpenes, which do not seem to have been detected here. Elaborate a bit on the probability of dealing with propolis here, or delete the mention.

We thank the reviewer for this comment. After conducting a literature review, it is clear that the lipid profile of propolis differs from beeswax and the archaeological samples. As such, mention of propolis in the discussion has been removed.

Discussion part, in particular l. 363-374: there is very little mention of the concomitant presence of fats, especially animal fats in most of the vessels where beeswax was detected. This is a fundamental aspect to consider when discussing the function of the containers. Please expand.

We thank the reviewer for this comment. Where relevant, we have expanded in the discussion on the relationship between beeswax and the plant and animal lipids identified. The pairing of the various signatures in particular may relate to unique recipes that combined unfiltered honey with other food products.

Still in discussion: from Figure 7, a clear distinction can be seen between BB-21 L2, whose beeswax profile is particularly well preserved, and the other 3 samples, whose profile is much more altered. Without ruling out the possibility of differential degradation, is it possible to discuss a potential different use of beeswax in these two groups of ceramics?

We thank the reviewer for this comment. The lipid profile of BB-21 L2 is unique from the other vessels for two reasons. First, the lipid yield in the solvent extract is significantly higher than the other samples. Second, in the acid fraction, dicarboxylic acids were identified. The prior more likely relates to intensive vessel use rather than superior preservation because in general, the lipid preservation from Tel Bene Beraq (South) is poor. The latter suggests that the beeswax was used alongside a specific plant product that cannot be presently identified. Most likely, the clear beeswax signature relates to a more intensive and task specific utilisation of this jar. This suggestion has now been added to the discussion.

1. 391-401: it cannot be excluded that the beehive products were exploited in perishable containers rather than in ceramics. Please include this hypothesis to the discussion.

We thank the reviewer for this comment and added this hypothesis into the discussion. Organic containers most frequently have been cited as manmade beehives.

Chromatograms of the extractions of the bound fraction and of all control samples are not presented in the article. To meet the publication requirements of the journal, they should be a second piece of Supplementary Information.

We thank the reviewer for this comment and fully agree. The relevant chromatograms have now been added to an additional supplementary material.

References:

Becker, Lillian C., Wilma F. Bergfeld, Donald V. Belsito, Ronald A. Hill, Curtis D. Klaassen, James G. Marks, Ronald C. Shank, Thomas J. Slaga, Paul W. Snyder, F. Alan Andersen, 2010. Final report of the amended safety assessment of myristic acid and its salts and esters as used in cosmetics. *International journal of toxicology* 29, 162S-186S

Casanova, E., Arbogast, R.-M., Denaire, A., Jeunesse, C., Lefranc, P., Evershed, R.P., 2020. Spatial and temporal disparities in human subsistence in the Neolithic Rhineland gateway. *J. Archaeol. Sci.* 122, 105215. <https://doi.org/10.1016/j.jas.2020.105215>

Drieu, L., Lucquin, A., Cassard, L., Sorin, S., Craig, O.E., Binder, D., Regert, M., 2021. A Neolithic without dairy? Chemical evidence from the content of ceramics from the Pendimoun rockshelter (Castellar, France, 5750-5150 BCE). *J. Archaeol. Sci. Rep.* 35, 102682. <https://doi.org/10.1016/j.jasrep.2020.102682>

Heron, C., Craig, O.E., Luquin, A., Steele, V.J., Thompson, A., Piličiauskas, G., 2015. Cooking fish and drinking milk? Patterns in pottery use in the southeastern Baltic, 3300-2400 Cal BC. *J. Archaeol. Sci.* 63, 33–43. <https://doi.org/10.1016/j.jas.2015.08.002>

Rageot, M., Lepère, C., Henry, A., Binder, D., Davtian, G., Filippi, J.-J., Fernandez, X., Guilaine, J., Jallet, F., Radi, G., Thirault, E., Terradas, X., Regert, M., 2021. Management systems of adhesive materials throughout the Neolithic in the North-West Mediterranean. *J. Archaeol. Sci.* 126, 105309. <https://doi.org/10.1016/j.jas.2020.105309>

Regert, M., Colinart, S., Degrand, L., Decavallas, O., 2001. Chemical alteration and use of beeswax through time: accelerated ageing tests and analysis of archaeological samples from various environmental contexts. *Archaeometry* 43, 549–569. <https://doi.org/10.1111/1475-4754.00036>

Stojanovski, D., Živaljević, I., Dimitrijević, V., Dunne, J., Evershed, R.P., Balasse, M., Dowle, A., Hendy, J., McGrath, K., Fischer, R., Speller, C., Jovanović, J., Casanova, E., Knowles, T., Balj, L., Naumov, G., Putica, A., Starović, A., Stefanović, S., 2020. Living off the land: Terrestrial-based diet and dairying in the farming communities of the Neolithic Balkans. *PLOS ONE* 15, e0237608. <https://doi.org/10.1371/journal.pone.0237608>

Whelton, H.L., Hammann, S., Cramp, L.J.E., Dunne, J., Roffet-Salque, M., Evershed, R.P., 2021. A call for caution in the analysis of lipids and other small biomolecules from archaeological contexts. *J. Archaeol. Sci.* 132, 105397. <https://doi.org/10.1016/j.jas.2021.105397>

The suggested references have been reviewed and added when relevant to the manuscript.

Reviewer 2:

This manuscript presents the evidence for the presence of beeswax in sherds from the Southern Levant using organic residue analyses. While the identification of beeswax in the samples seem rather robust (more on this below), the paper is discussing rather uncritically some previous identification of “beeswax” in the Levantine record. A discussion regarding previous identification should be included in the paper and the necessary changes made throughout to reflect the erroneous identification of beeswax.

I recommend this paper to be published with minor corrections (detailed below).

Ref [31] (Namdar et al. 2009, JAS) identifies putative beeswax in cornets from the Chalcolithic based on the distribution of n-alkanes. However, the sole presence of n-alkanes (when fatty acids / alcohols / wax esters are absent) cannot be seen as a proof for the presence of beeswax. Whelton et al. 2021 (JAS) have written about this recently in their “call for caution”. This argument should be detailed in the manuscript as many of the published literature about “beeswax” in the Levant is not identifying beeswax in the way it should be. Those finds (such as ref 34 at line 62 or ref 35 at line 67; but there are others) should not be included in the list of sites with evidence for beeswax. This comment about n-alkanes leads me to the structure of the paper. The introduction exposes the evidence for the exploitation of bee products in the region, including from organic residue analyses. There is a need of a paragraph in the introduction discussing the suite of compounds that are characteristic of beeswax. This information comes right at the end – with the study of fresh beeswax – but this is too late, as the reader cannot relate the archaeological lipid assemblage to what is expected for beeswax and degraded beeswax.

We thank the reviewer for all these comments. We agree that the many of the prior beeswax identification in the Levant are problematic because they are based on just the *n*-alkane profiles. However, contradicting these suggestions is difficult because the articles do not publish the entire chromatograms, and therefore, the real critique is the lack of data to allow one to draw an informed conclusion. Instead of removing these references, they are cited with more caution and mention of their flaws.

To avoid any confusion in this paper about how beeswax was identified, the qualifiers based on published beeswax chromatograms were added to the second paragraph of the

introduction. It is now presented only in a condensed version in the results section to aid in confirming the identification of beeswax in the archaeological samples.

Affiliations: to be double-checked as there are 8 listed affiliations but only 7 numbers by the authors' names.

We thank the reviewer for this comment. This mistake has been fixed.

Abstract: should be reworked based on my comments about the manuscript.

We thank the reviewer for this comment. The abstract has been modified after incorporating the comments of the two reviewers.

l. 44-45 “although the proportion is potentially biased by preservation conditions and sample size”. This sentence is unclear – please rephrase.

The sentence has been rephrased to reflect that the percent of identified beeswax may be influenced by the number and types of vessels sampled as well as the preservation conditions offered by climate and sediment characteristic of each site.

l. 75. “Beeswax is rare” – this statement should be supported by numbers. How many sherds analysed? How many sherds with appreciable amount of lipids? How many beeswax residues? Due to the poor preservation of lipids in the region, residues with significant amount of lipids are rare, but within that beeswax are rather common (?).

Based on table 1, 467 sherds have been analysed of which the authors have lipid data for 457. Around 33 have > 5 ug/g of lipids (recovery rate of 7%). Beeswax has been evidenced in 14 vessels (this number will be lower when sherds with n-alkanes only will be removed). Beeswax is thus recorded in a whopping 42% of sherds!

We agree with the reviewer that if you rely solely on the TLEs based on solvent extraction, the percentage of beeswax is rather high. Ca. 35% of all internally solvent extracted samples contain evidence of beeswax, and if you exclude the results of Namdar as you suggested, it is only 10%.

However, the frequency of beeswax can be biased by the extraction technique and how the TLE is calculated. In this study, the recovery rate is based on the solvent extracts; recent

studies though calculate the TLE and recovery rate based on the direct acidified methanol extraction and only then solvent extract additional ceramic powder. The percentage of beeswax in these samples then is the N sherds where it was identified in the solvent extract out of the N sherds with over 5 µg g⁻¹ of lipids recovered from direct acidified methanol extraction (see for example the paper by Dunne et al. 2021 on West Africa). This creates a lower percentage of beeswax because the number of sherds with significant lipid yields is higher. If we were to base the TLE and recovery rate on the combined lipid yields from solvent extraction and saponification (which should be the equivalent of the direct acidified methanol extraction results), then for the samples analysed as part of this study, the frequency of beeswax would be less than 5%.

Rather than support the rarity of beeswax by adding in frequencies based on lipid recovery (that may be biased by how it is calculated), we have reinforced the rarity by emphasizing the low number of sites where beeswax was found. Beeswax was identified in 3/19 sites analysed in prior studies, following the results of Namdar, and 7/30 including those analysed by the authors of this study.

l. 101. Worth mentioning in this paragraph the number of sherds that were analysed in this study; with number of sherds for each vessel type and each sites. That would allow the reader to grasp the full coverage of the study. Crusts should be mentioned here too.

We thank the reviewer for this comment. Also following the advice of Reviewer 1, we have chosen to include this data through an additional table. Sherd and type specific data will however be included in a future paper.

l. 111. More details is needed about the internal and external surfaces, as this does not seem to be a standard protocol (no literature relating to the sampling method is cited). What depth? What area? How do they relate one to another? The authors seem to have analysed some crusts too – what is the sampling protocol for those?

We thank the reviewer for this comment. Additional details have been added about how many grams were sampled and the depth that was drilled.

l. 114. “Lipids from the external ceramic surface or soil samples were analysed as controls”. Why? Add “to check for xxx”.

We thank the reviewer for this comment and have added further details on this. They were analysed to check for post-depositional contamination and contamination from handling.

l. 118. “to ensure that the ‘bound’ lipids were extracted without hydrolysis of the esterified lipids”. Surely the main reason of using a solvent-extraction is to not hydrolyse hydrolyseable (WE and HWE) lipids? Could be rephrased “to ensure that solvent-soluble lipids were extracted without hydrolysis of the esterified lipids”.

We thank the reviewer for this comment and have changed the phrasing of this sentence accordingly.

l. 120. “Lipids were extracted three times”. Details are needed here regarding length of time for the sonication step.

This has been expanded on. Samples were sonicated for 15 minutes and centrifuged for 10 minutes at 3000 rpm.

l. 125. “Hexatriacontane (C₃₆ n-alkane; 98%, Sigma-Aldrich) was measured as an internal or external standard for TLE quantification”. This does not make sense. Are the authors adding a known amount (how much?) of IS in their lipid extracts for quantification? Or are the extracts quantified using a known amount of compounds? Same comment for l. 133.

We thank the reviewer for this comment. How the standard was used has been further clarified. Upon noticing that hexatriacontane frequently co-eluted with other molecules, the methodology was changed, and it was run systematically in the GC-MS sequence after every four samples to allow for a more accurate quantification.

l.153. “Lipid preservation was challenging”. This is not correct. Do the authors mean “low”?

We agree and have changed the text accordingly.

l. 158. “which may relate to spillage or the application of a vessel treatment”. It is worth elaborating on this idea a bit further. What would lead to absorbed residues being only deposited only in the outer surface? Cite some literature here too.

We thank the reviewer for this comment and have expanded on this point. These external residues most likely relate to a post-firing treatment rather than post-depositional

contamination, lipid migration from the vessel interior or spillage because there are insignificant lipid yields internally.

l. 161. “Among these” – please be more specific, as the previous paragraph talks about multiple extracts.

We thank the reviewer for this comment and have specified that this relates to the internally and externally solvent extracted vessels.

l. 167. As explained earlier – this should be amended as it refers to the identification of beeswax based solely on n-alkanes.

We agree with the reviewer and have amended the text accordingly.

Table 1 is rather confusing (and should go into the SI as this is a repeat from the GC/MS chromatograms and the histograms from Fig 7). That would be good to visually group the 2 samples that are coming from the same sherd (outer/inner surface). “Monopalmitin” should read “monopalmitin”. Should BB-12 and BB-21 be the same sample? What are Layers? What is the difference between “TSF18-57 In L1” and “TSF18-57 In L2”? This should be explained in the Methods section.

We have clarified table 1 (now table 2) accordingly and have chosen to leave it in the manuscript. BB-12 is a soil sample that relates contextually and spatially to the ceramic vessel BB-21, but because of the excavation methodology, these were collected separately. To avoid this confusion, the soil sample has been relabeled as BB-21s.

Comments about the four sherds with beeswax lipid assemblage:

NY-12 (from line 176) Cholesterol is known to oxidise rapidly in presence of fatty acids and clay (Hamman et al. 2018, Tetrahedron Letters). Given the very poor preservation of lipids at the site, the detection of cholesterol in that sample is rather suspicious – this could potentially indicate modern contamination.

We thank the reviewer for this comment and have already acknowledged in the text that cholesterol may relate to contamination. This point has been reemphasised.

Unsaturated fatty acids (C16:1, C18:1 and C20:1) are interpreted as arising from plant oils. The C18:1 is very likely to arise from carcass fats (very abundant in fresh carcass fats). The C16:1 and C20:1 are not labelled in the chromatogram (Fig 3) – are they abundant? It seems rather overstretched for me to interpret those as arising from plant oils as they are also present in other foodstuffs (eg modern salmon). What is the interpretation for K31 and K33? Even-chain alkanes are not detected in modern beeswax, so their presence here is rather puzzling (and would suggest contamination). I would argue that there is only weak evidence for beeswax in this sample (lack for LCFAs, alkanes, weak evidence for WE etc) – the interpretation should be tentative.

We thank the reviewer for this comment. C_{18:1} may originate from plant oil and animal fat, and this was already stated in the text. The remaining unsaturated fatty acids as shown in Table 1 (now Table 2) were found in the acid fraction, and these are displayed in Supplementary material 2. Fish was not considered as a potential origin because, while targeted fish biomarker analysis is underway, the faunal record shows a paucity of fish remains despite intensive sifting. Further, fish are characterised by multiple isomers of unsaturated fatty acids, while only one isomer of C_{20:1} was identified. An explanation on the ketones is given in the discussion in relation to vessel function. The even-numbered *n*-alkanes do suggest an additional lipid input, and this was acknowledged in the results section.

We acknowledge that based exclusively on the solvent extract, the identification of beeswax is tenuous. However, when the results are combined with the saponified acid fraction, the identification is clearer. Long-chain even-numbered saturated fatty acids were identified abundantly as shown in Table 1 (now Table 2). Nonetheless, we have rephrased the text to present this sample with greater caution.

TSF-18-57 (from line 202) This sample has a much stronger signature of degraded beeswax. Again, I am sceptical about the interpretation of plant oils re: C_{18:n} fatty acids.

We thank the reviewer for this comment. Both interpretations (plant oil and animal fat) are now presented.

BB-21 (from line 226) Good evidence there too, although the alkane distribution is rather peculiar. The distribution of the WE/HWE is very characteristic and well-preserved.

TS-4 (from line 249) Rather high even-numbered alkanes in this sample. The WE distribution is not very characteristic and well-preserved. The presence of beeswax is only tentative.

We thank the reviewer for this comment and have rephrased the text so that the identification is presented more tenuously.

The chromatograms (Fig. 3-6) are rather difficult to read, as the amount of labels makes the chromatogram very busy and because of the thickness of the line indicating peaks of small size (eg for cholesterol, C15:0 and A16 in Fig 3) – it looks like the peak is much bigger than it is. I would suggest to use symbols and colours to indicate compound classes – with symbols just above the peak – that allows to see at first glance the distribution of compounds within compound classes (eg see Roffet-Salque et al. 2015, Nature).

We thank the reviewer for this comment. To clarify the figures, the lines for the small peaks have been changed in color with an arrow added at the end.

1. 259. The lipid distribution of beeswax is species-specific (see Aichholz et al. 1999, J of Chromatography A) – so it is particularly important to provide details regarding the species (I assume *A. mellifera*).

We thank the reviewer for this comment and have specified from which type of bees the reference beeswax originated.

1. 286 High molecular weight WE are usually well-preserved, with low-molecular weight WE degrading first. Could their absence in the extract be due to the GC-MS setup?

We thank the reviewer for this comment. The machine settings are similar to those used in other studies that identified beeswax, so it is unlikely that this affected the wax esters recovered.

1. 298 Four out of 247 vessels contained beeswax. This is interesting but lipid preservation is rather low. How many sherds with appreciable amount of lipids? You should compare that number to the number of beeswax.

We thank the reviewer for this comment. As noted earlier, calculating the frequency of beeswax is problematic depending on how the recovery rate is defined. Nonetheless, we have

expanded the text to also show the frequency of beeswax in relation to the number of samples with over 5 µg g⁻¹ of lipid in the internal or external solvent extract.

l. 306 “calculated TLE”. What does this mean? The lipid concentration was not calculated? (same in table 1 caption).

We thank the reviewer for this comment and have clarified accordingly.

l. 317 Cornets here again.

The identification of beeswax in cornets has been cited more tenuously.

l. 338 Evidence for waterproofing collared flasks with beeswax has also be shown by Salque et al 2013 (Nature).

We thank the reviewer for pointing this out and have added in the reference.

l. 354 The authors use the presence of beeswax as potential evidence for propolis. What is the chemical composition of propolis? Does it fit with this interpretation?

We thank the reviewer for this comment. After reading more on the lipid profile of propolis, we agree that the residue cannot result from this. As such, mention of propolis has been removed from the discussion.

l. 379 Misidentification of beeswax in cornets here again.

We agree with the reviewer that the identification of beeswax in the cornets is problematic, but we cannot just ignore the publication because it is archaeologically highly relevant. The use of cornets has been presented more cautiously

l. 391 See my comment for l. 75. Results should be reported against number of sherds with archaeological residues, rather than total number of sherds.

The percentages based on the recovery rates have been added. We however believe that the true frequency is observed by the low number of sites with evidence for beeswax and the low number of sherds with evidence for beeswax at each of these sites.

l. 400 For Tell Sabi Abyad, 287 vessels were sampled (the citation to ref 75 is erroneous and should be ref 95) but only 41 led to significant amount of archaeological lipids. So no beeswax in 41 sherds with lipids.

We thank the reviewer for this comment and have clarified this in the text.

l. 403 Not sure “easily” is the right term for exploiting wild bees, given the difficulties honey hunters usually face to gather beeswax and honey.

We thank the reviewer for this comment and have removed the word easily.

l. 405 I see that the discussion is building on the low frequency of beeswax in the Levant, however, as I said earlier, this is hampered by the lack of residues. The recovery rate based on sherds with archaeological lipids is much higher, and probably not far off from other regions... I would advise the discussion section to be re-phrased to take this into account.

We thank the reviewer for this comment. As we already noted, the frequency of beeswax is problematic because of the low lipid yield and reliance on just the solvent extraction to calculate the recovery rate. The frequency is now presented with greater transparency and increased caution. However, even without comparing to chronologically parallel European sites (with improved lipid preservation), we do stand by the suggestion that the use of beeswax was low, especially when compared to periods post-dating the Neolithic and Chalcolithic in the region.

Figure 1. Panel A is an update from a figure from ref 17. Panel B is very relevant to the paper. However, it is rather misleading as many of the sites did not lead sherds with significant amount of lipids. This figure does thus not allow to really see what is happening in the region (given that the authors are arguing for low intensity). This information could be added – nb of sherds analysed, nb of sherds with lipids, nb of sherds with beeswax (or percentages) maybe using pie charts? That way Table 2 could move to SI.

We thank the reviewer for this comment. We agree that it would be helpful to have this data in the figure, but because of spatial limitations and overcrowding, the data was left in table format and was not added to the figure.

References. Please check the references from misspelling etc as there seem to be some issues. Eg Ref 70 should read “Eglinton”

We thank the reviewer for pointing out this mistake. The references have been reviewed and any errors have been corrected.